# Fast-in-Slow: A Dual-System VLA Model Unifying Fast Manipulation within Slow Reasoning

**Hao Chen**[*1,2] **Jiaming Liu**[*,†2] **Chenyang Gu**[*2,4] **Zhuoyang Liu**[*2] **Renrui Zhang**[†1] **Xiaoqi Li**[2]
**Xiao He**[3] **Yandong Guo**[3] **Chi-Wing Fu**[1] **Shanghang Zhang**[2,4 ✉] **Pheng-Ann Heng**[1]

[1]The Chinese University of Hong Kong [2]State Key Laboratory of Multimedia Information Processing,
School of Computer Science, Peking University
[3]AI[2]Robotics [4]Beijing Academy of Artificial Intelligence (BAAI)

## Abstract

Generalized policy and execution efficiency constitute the two critical challenges in robotic manipulation. While recent foundation policies benefit from the common-sense reasoning capabilities of internet-scale pretrained vision-language models (VLMs), they often suffer from low execution frequency. To mitigate this dilemma, dual-system approaches have been proposed to leverage a VLM-based System 2 module for handling high-level decision-making, and a separate System 1 action module for ensuring real-time control. However, existing designs maintain both systems as separate models, limiting System 1 from fully leveraging the rich pretrained knowledge from the VLM-based System 2. In this work, we propose Fast-in-Slow (FiS), a unified dual-system vision-language-action (VLA) model that embeds the System 1 execution module within the VLM-based System 2 by partially sharing parameters. This innovative paradigm not only enables high-frequency execution in System 1, but also facilitates coordination between multimodal reasoning and execution components within a single foundation model of System 2. Given their fundamentally distinct roles within FiS-VLA, we design the two systems to incorporate heterogeneous modality inputs alongside asynchronous operating frequencies, enabling both fast and precise manipulation. To enable coordination between the two systems, a dual-aware co-training strategy is proposed that equips System 1 with action generation capabilities while preserving System 2's contextual understanding to provide stable latent conditions for System 1. For evaluation, FiS-VLA outperforms previous state-of-the-art methods by 8% in simulation and 11% in real-world tasks in terms of average success rate, while achieving a 117.7 Hz control frequency with action chunk set to eight. **Project web page:** fast-in-slow.github.io.

## 1 Introduction

The undamental objective of robotic manipulation learning [3, 4, 5, 6] is to convert real-world sensory data and human instructions into precise control signals. Simultaneously, enabling robots to execute a broad spectrum of tasks while adapting to variations in objects and environments remains the core challenge. Recently, some works [7, 8, 9, 10, 11, 12] have sought to leverage the pretrained knowledge of foundational vision–language-models (VLMs) [13, 14, 15, 16, 17, 18] to enable generalized manipulation by fine-tuning these models on robotic datasets [19, 20], giving rise to the vision-language-action (VLA) models. However, these methods, with their billion-scale parameters and autoregressive action generation, lead to low operating frequencies, which constrain responsive closed-loop control and hinder real-world application.

---

*Equal contribution, †Project lead, ✉Corresponding author.

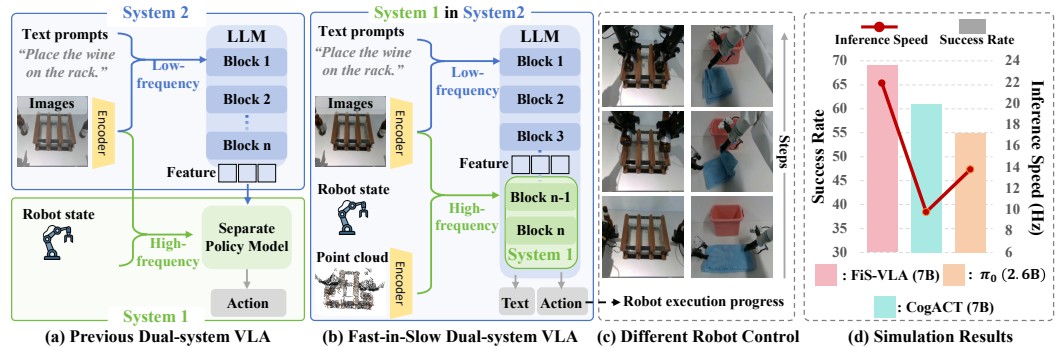

Figure 1: **Overview of FiS-VLA.** (a) Unlike previous dual-system VLA methods [1, 2] that attach a separate policy head as System 1, FiS-VLA (b) repurposes the final transformer blocks of an intact VLM as System 1, while retaining the full model for System 2 reasoning. Under this paradigm, FiS-VLA achieves superior performance and high-frequency control, as shown in (c) and (d).

Drawing inspiration from Kahneman's dual-system theory [21] that *"System 1 is fast, intuitive, and unconscious, while System 2 is slow, logical, and involves deliberate reasoning"*, recent works have explored incorporating dual-system design into VLA models. Most recent end-to-end approaches [22, 23, 24] leverage VLM as System 2 for high-level feature extraction, while appending an additional policy head as System 1 to transform VLM outputs into executable action poses. Building on a similar architecture, methods such as [2, 1, 25] design dual-system frameworks with asynchronous operating frequencies, further clarifying the distinct roles of System 1 and System 2. While these methods improve execution efficiency, their System 1, as a lightweight separate model, lacks internet-scale pretrained knowledge and depends solely on feature representations extracted by System 2, thus failing to fully leverage the reasoning capabilities within System 2's VLM. Considering these limitations, and motivated by the functional abstraction of Kahneman's dual-system theory, we raise a question: *"If a VLM model serves as the central decision-making module of the robot, can it integrate System 1 and System 2 processes to enable coordinated reasoning and execution?"*

To this end, we propose Fast-in-Slow (FiS), a VLA foundation model that integrates the fast execution capabilities of System 1 into a pretrained VLM, while preserving its inherent System 2 contextual understanding and generation capabilities. As shown in Figure 1, unlike prior dual-system VLA approaches [1, 25] that attach System 2 with an independent policy model as System 1, FiS-VLA repurposes the final transformer blocks of System 2 into a high-efficiency execution module, serving as System 1. Under this dual-system paradigm, FiS-VLA enables seamless coordination between the two systems, as both are derived from the same foundation model without altering its connectivity structure. Since System 2 handles understanding and reasoning while System 1 focuses on rapid action execution, we design the two systems with heterogeneous modality inputs and asynchronous operating frequencies. For System 2, it operates at a lower frequency, processing 2D observations and language instructions into multimodal latent representations that guide System 1's actuation. For System 1, we systematically investigate the impact of various high-frequency inputs for robot control, including the robot state, 2D images, and 3D point clouds. Notably, since 3D geometric information is critical for precise manipulation [26, 27], we utilize a fast 3D embedding strategy that tokenizes point clouds [28] and processes them through a shared vision encoder to extract spatial features, which directly condition the System 1 for geometry-aware interaction.

To jointly optimize the reasoning and execution components in FiS-VLA, we introduce a dual-aware co-training strategy. For the execution component (System 1), we adopt the probabilistic and continuous nature of diffusion modeling [3, 29, 30] by injecting noised actions as latent vectors into the embedding space of System 1 to learn action generation. For the multimodal comprehension component (System 2), we exploit an autoregressive next-token prediction objective to maintain its discrete action generation or high-level language planning capabilities and preserve the overall coherence and integrity of System 2. Under this co-training approach, FiS-VLA first undergoes large-scale pretraining on open-source robotic datasets [31, 20, 32] comprising more than 860K trajectories. It is then fine-tuned on high-quality, self-collected real-world and simulation data [33]. In both real-world and simulated experiments, FiS-VLA achieves state-of-the-art (SOTA) manipulation performance. Meanwhile, FiS-VLA demonstrates strong generalization to unseen objects, complex

backgrounds, and diverse lighting conditions, regardless of the robot type. With a 1:4 operating frequency ratio between System 2 and System 1, FiS-VLA achieves a 117.7 Hz control frequency on an NVIDIA 4090 GPU with action chunk set to eight. In summary, our contributions are as follows:

- We propose Fast-in-Slow (FiS), a unified dual-system VLA model that embeds System 1 execution within a pretrained VLM while preserving its inherent System 2 reasoning capabilities, thereby enabling seamless coordination between both systems.

- Given that System 2 and System 1 serve fundamentally distinct roles within FiS-VLA, we systematically design them with heterogeneous modality inputs and asynchronous operating frequencies, enabling both fast and precise manipulation.

- We propose a dual-aware co-training strategy to jointly optimize System 2 and System 1 in FiS-VLA. Our model demonstrates SOTA performance in both single-arm simulation and dual-arm real-world experiments, while maintaining a high execution frequency.

## 2 Related Work

**Vision-language-action models.** Early approaches for robot manipulation primarily relied on reinforcement learning with reward functions derived from proprioceptive signals [34, 35, 36, 37], as well as imitation learning based on visual observations [38, 39, 40, 3]. More recently, increasing attention has been directed toward integrating vision-language models (VLMs) into robotic systems [41, 42, 43, 44, 12, 11, 45, 46, 30], leading to the emergence of Vision-Language-Action (VLA) models. These models leverage the reasoning capabilities of VLMs to directly predict low-level SE(3) poses for manipulation tasks. A common approach among prior works [47, 48, 7, 49] involves autoregressive next-token prediction to generate action sequences. However, such methods often suffer from action discontinuities and low execution frequency. To mitigate this, some VLA models [50, 51, 52, 53] incorporate a policy head to enable continuous action prediction. Recent studies [27, 54, 55, 56, 57] emphasize the value of 3D spatial information for robotic manipulation, making 3D observations a common strategy to boost spatial understanding and accuracy. Furthermore, it has been demonstrated [7, 22, 30, 10] that pretraining VLA models on large-scale robotic datasets [19, 20, 32, 58] can significantly improve their generalization capability. However, these VLA methods commonly suffer from low action generation frequency and still lack the adaptability to adjust their behavior with low latency in response to dynamic task requirements.

**Dual-system design in VLA.** To improve the execution frequency of VLA models, some recent methods split the framework into two systems. System 2 is responsible for high-level task reasoning, while System 1 focuses on low-level action generation. Methods such as [23, 22, 59, 24] adopt a VLM as System 2 to produce a latent feature. This latent feature is then used as a condition for a diffusion-based action head (System 1). However, in these methods, System 1 and System 2 typically operate at the same frequency, limiting System 1's potential for high-frequency action generation. We refer to this design as a synchronous dual-system architecture. Additionally, [60, 1, 61, 2, 25] adopt a similar architecture but operate System 2 and System 1 at different frequencies. This asynchronous design further improves the overall action generation frequency of the VLA model. Moreover, some methods [62, 63] attempt to incorporate subtask decomposition within a synchronous dual-system architecture to enhance task planning. Nevertheless, all of these methods introduce a new untrained System 1 policy head and rely solely on features extracted by the System 2 VLM, thus failing to fully leverage the VLM's pretrained knowledge and reasoning capabilities [64]. In this work, we propose FiS-VLA, an asynchronous architecture that integrates a System 1 execution module into a System 2 VLM, enabling seamless coordination between the two systems within a single pretrained model.

## 3 Fast-in-Slow Dual-System VLA

In this section, we introduce our proposed FiS-VLA framework, as shown in Figure 2. We begin in section 3.1 with a formal problem formulation. In section 3.2, we describe the overall architecture of FiS-VLA. The core idea of our approach is to retain an intact VLM for System 2 reasoning, while repurposing its final transformer blocks into a System 1 execution module. This design constructs System 1 not as an independently injected module, but as a component that inherits the VLM's pretrained knowledge and maintains coherent understanding of the intermediate reasoning outputs of System 2, while meeting the low-latency demands of real-time control. In Section 3.3, we present the motivation and detail the mechanisms for designing the two systems to operate at

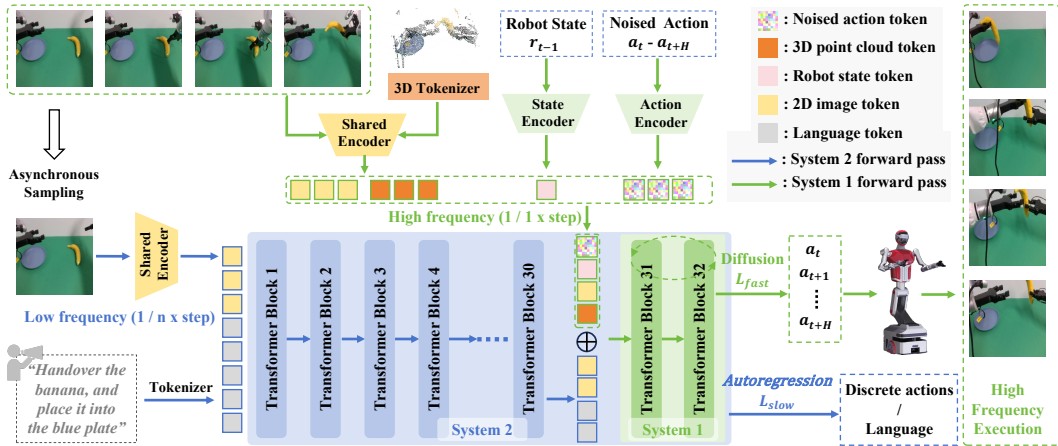

Figure 2: **Framework of FiS-VLA.** FiS-VLA leverages an intact VLM for System 2 reasoning while repurposing the final transformer blocks of the LLM for System 1 execution module. System 2 handles low-frequency inputs such as 2D images and language instructions and produces intermediate latent features that serve as conditioning information for System 1. Instead of being conditioned solely on these periodically updated high-level representations, System 1 processes high-frequency inputs including 3D point clouds, 2D images, and robot states to produce stable and responsive actions. For joint optimization, we introduce a dual-aware co-training strategy that combines a diffusion denoising objective with an autoregressive objective which enables FiS-VLA to support fast action generation while retaining System 2's multimodal reasoning capabilities.

asynchronous frequencies with different input modalities. Finally, in Section 3.4, we show our dual-aware co-training strategy that jointly optimizes both systems. Within this training framework, FiS-VLA leverages System 1 for continuous action generation while employing System 2 for discrete action or language generation.

## 3.1 Problem Formulation

Following [23, 22], VLA models typically learn robotic control policies through imitation learning on heterogeneous demonstration datasets $\mathcal{D}$. The training objective is to maximize the likelihood of generating temporally extended action sequences $a_{t:t+H}$, conditioned on multimodal observations $o_{t-1}$ and language instructions $l$. In this work, we construct comprehensive observations, including the robot state, multi-view images, and 3D point clouds. Formally, given the policy model $\pi_\theta$, this corresponds to the optimization problem:

$$\max_\theta \ \mathbb{E}_{(a_{t:t+H}, o_{t-1}, l) \sim \mathcal{D}} \left[ \log \pi_\theta(a_{t:t+H} \mid o_{t-1}, l) \right].$$

The action $a$ can represent different control spaces and control modes. In this work, we employ 7-DoF end-effector pose control for the single-arm Franka Panda robot in simulation, consisting of 3-DoF for relative positional offsets ($[\Delta x, \Delta y, \Delta z] \in \mathbb{R}^3$), 3-DoF for rotation (represented as Euler angles, $\in \mathbb{R}^3$), and 1-DoF for gripper state (open/closed, $\in \mathbb{R}^1$). For real-world experiments, to validate our model's robustness across different robot embodiments and control modes, we employ 14-DoF control on the AgileX and 16-DoF control on the AlphaBot dual-arm robots, under the end-effector pose control and joint position control, respectively.

## 3.2 FiS-VLA Architecture

We begin by presenting an overview of the FiS-VLA architecture, as shown in Figure 2. Similar to prior VLA methods [7, 22], FiS-VLA inherits the base architecture and initializes pretrained parameters from Prismatic VLMs [16]. The model primarily consists of a vision encoder and a LLM, with an additional lightweight 3D tokenizer introduced to efficiently process point cloud inputs.

**Vision encoder.** We employ both SigLIP [65] and DINOv2 [66] to jointly extract visual representations that capture high-level semantic features and local spatial details. Specifically, for each

input image, we first resize it to 224×224 pixels. The image is then processed by both encoders, yielding two distinct feature representations $f^{\text{SigLIP}} \in \mathbb{R}^{N_v \times 1024}$ and $f^{\text{DINO}} \in \mathbb{R}^{N_v \times 1152}$, where $N_v$ represents the token dimension. These two features are concatenated along the channel dimension, resulting in a unified visual embedding for further processing.

**Point cloud encoder.** To investigate the impact of 3D geometric information on robotic manipulation, we incorporate point cloud data $\mathcal{P} = \{\mathbf{p}_i \in \mathbb{R}^3\}_{i=1}^{N_p}$, which is derived from a single-view depth map using camera intrinsics and extrinsics. $N_p$ denotes the number of points. Unlike some approaches [54, 27] that directly process point clouds with newly injected 3D encoders, our method first transforms the point cloud into high-dimensional tokens using a lightweight 3D tokenizer [67]. Specifically, the 3D tokenizer consists of three blocks, each containing farthest point sampling [68] for downsampling, the k-nearest neighbors algorithm for local aggregation, and a learnable linear layer for feature encoding. The tokenized representation is then processed by our shared vision encoder to extract local spatial features, following [28]. This design offers two key advantages: first, it effectively projects 3D information into the LLM's embedding space by leveraging the vision encoder of the pretrained VLM with vision-language alignment capabilities; and second, it avoids obvious parameter increase and maintains computational efficiency.

**LLM backbone.** The 7B LLaMA2 [69] model is adopted as the LLM backbone for FiS-VLA. LLaMA2 is a decoder-only transformer architecture consisting of 32 blocks, where the input and output of each block can be viewed as high-dimensional representations of a token sequence. Previous works [24, 70] find that, in VLA models, leveraging intermediate LLM representations instead of the final layer for action generation improves downstream policy success rates without degrading multimodal representation quality. Therefore, we repurpose the final few blocks of the LLM for System 1 to condition on intermediate latent features from System 2, enabling efficient, low-latency responses. To ensure that System 2 maintains its full reasoning capability, we incorporate the complete LLM as its core component, forming a *"fast system within slow system"* architecture that balances rapid action generation with deep contextual reasoning.

**MLP components.** To further clarify the FiS-VLA architecture, we describe the remaining auxiliary components, all of which are implemented as MLPs. First, a pretrained vision-language projector is employed to map 2D and 3D features into the LLM's textual embedding space, which is initialized from the pretrained VLM. In parallel, the robot proprioceptive state is encoded using a state encoder. Given that we adopt diffusion-based action generation for System 1, two additional MLPs are incorporated to project the timestep and noised actions as continuous vectors.

### 3.3 Dual-System Coordination

**Asynchronous frequency design.** FiS-VLA is structured into two components: a slow System 2 and a fast System 1, inspired by Kahneman's dual-system theory [21]. Since System 2 VLM with billion-scale parameters, it operates at a low frequency to perform high-level semantic understanding and contextual reasoning. In our framework, it interprets task-relevant visual observations and language instructions, and produces a comprehension output in the form of latent features from an intermediate block of the LLM. Building on previous action chunking methods [39, 3], the instruction and scene observation at time step $t$ can provide guidance for a future horizon of action steps ($a_{t:t+H}$). Consequently, System 2's intermediate output serves as a latent condition that temporally guides System 1's action generation across the following $H$ time steps. In contrast, System 1 focuses on generating executable actions in real time. At each time step, it leverages the most recent observation to generate actions, while being conditioned on the periodically updated high-level reasoning output from System 2. This behavior resembles intuitive and reactive responses, positioning System 1 as a high-frequency action generation module.

To achieve this, we investigate the coordination frequency between the two systems. A central question is that *"How many future action steps can be effectively guided by the intermediate comprehension output from System 2?"* We empirically explore the effect of varying horizon lengths (e.g., 1, 2, 4, ..., n) in the ablation study. This corresponds to setting the operating frequency ratio between System 2 and System 1 to 1:$n$, as our robot's hardware does not support deploying the two systems on separate GPUs for parallel inference, unlike the implementation in Helix [25]. While parallel inference can further improve model speed, we focus on the fundamental research question of identifying the optimal coordination ratio between two systems. In Figure 2, to ensure that System 1 can effectively interpret the latent conditions produced by System 2 from earlier horizon steps, we

employ asynchronous sampling during training to reduce the operating frequency of System 2. This encourages the System 1 execution module to maintain temporal consistency in task understanding.

**Heterogeneous modality input.** The two systems in FiS-VLA are designed to serve fundamentally distinct purposes. System 2 is responsible for high-level task understanding and scene reasoning, whereas System 1 is optimized for fast, reactive control. In line with these different objectives, we propose that each system should be provided with input modalities specifically tailored to its function. Since the System 2 VLM has undergone internet-scale pretraining on image-text paired data, we provide it with both language instructions and 2D visual observations to fully exploit high-level semantic reasoning capabilities. In contrast, System 1 is tasked with generating executable actions in real time, conditioned on a comprehensive representation of the robot's current environment. We carefully explore the information required for accurate and responsive control. First, System 1 must receive low-latency 2D images of the current scene. To enhance temporal consistency in closed-loop control, the robot's current state is also essential. Furthermore, since the robot must reason about spatial relationships and interact with complex spatial configurations, we additionally provide 3D point cloud data to support precise manipulation. Ultimately, the System 1 execution module integrates the three input modalities with the periodically updated latent feature from System 2, jointly serving as the conditioning context for diffusion-based action generation. Our experimental results confirm that each modality contributes meaningfully to the success of the manipulation tasks.

### 3.4 Training Objective and Recipe

**Dual-aware co-training strategy.** The core objective of FiS-VLA is to generate accurate and executable actions. To this end, we leverage the continuous nature of diffusion modeling, which typically yields more reliable actions than discrete prediction approaches [22, 23]. Given an initial action sequence $\tilde{a}$, we inject Gaussian noise $\eta \sim \mathcal{N}(0, I)$ at a randomly chosen timestep $\tau \sim \mathcal{U}(1, T)$, where $\tau \in \mathbb{Z}$ and $T = 100$. The forward process adds noise in closed form: $\tilde{a}_\tau = \sqrt{\beta_\tau}\tilde{a} + \sqrt{1 - \beta_\tau}\eta$, where $\beta_\tau$ denotes the noise scaling factor according to a predefined schedule [71]. To train System 1 $\pi_{\theta_f}$, we formulate the learning process as an optimization problem over the following objective:

$$\mathcal{L}_{\text{fast}} = \mathbb{E}_{\tau, c, \tilde{a}, \eta}\left[\|\eta - \pi_{\theta_f}(\sqrt{\beta_\tau}\tilde{a} + \sqrt{1 - \beta_\tau}\eta, c, \tau)\|^2\right], \tag{1}$$

where $c$ denotes the conditioning sources. In FiS-VLA, $c$ consists of two components: the low-frequency latent feature extracted from System 2, and the high-frequency input to System 1. Since the System 1 execution module is embedded within the System 2 VLM, exclusively training the model for diffusion-based action generation may lead to catastrophic forgetting of its autoregressive reasoning capability. To mitigate this issue, we propose a joint training objective to the entire VLA model that preserves System 2's reasoning ability by incorporating next token prediction with a cross-entropy loss. The autoregressive supervision signal can be either discrete actions [7, 49] or language-based plans [50, 8], depending on the construction of the robotic training data. As an example with discrete actions, we define the objective as follows:

$$\mathcal{L}_{\text{slow}} = -\sum_{i=1}^{D_t} \log P(\hat{a}_i \mid (context), \theta), \tag{2}$$

where $D_t$ represents the total length of discrete action tokens, $\hat{a}_i$ denotes the $i$-th ground-truth action action token, and $P(\hat{a}_i \mid context, \theta)$ is the probability predicted by the LLM given the input context and model parameters $\theta$ ($\theta_f \subseteq \theta$). Finally, we derive the overall training objective to update the FiS-VLA model.

$$\mathcal{L}_{\text{FiS-VLA}} = \mathcal{L}_{\text{fast}} + \mathcal{L}_{\text{slow}}. \tag{3}$$

**Pretraining recipe.** Prior to pretraining FiS-VLA, we initialize the model with parameters from a pretrained VLM [16], following the method established in [7, 22]. We curated a specialized pretraining dataset by carefully processing and filtering large-scale cross-embodiment datasets including Open X-Embodiment [19], DROID [20], ROBOMIND [32], and so on. As detailed in Appendix A, this dataset comprises over 860K trajectory samples. FiS-VLA was trained on this dataset for five epochs, with both system inputs consisting solely of a single image as observation. Since the pretraining data contains no subgoal-level language instructions, we initially supervise System 2 using discrete action sequences. During fine-tuning, we enhance the System 2 objective with additional language supervision by incorporating manually annotated sub-task plans and applying automated augmentation.

Table 1: **Comparison of FiS-VLA and baselines on RLBench.** All methods are trained in the multi-task setting [73], and we report success rates (S.R.) based on the evaluation criteria defined in RLBench. Inference speed is evaluated on an NVIDIA 4090 GPU with action chunk set to one.

| Models | Close box | Close laptop lid | Toilet seat down | Sweep to dustpan | Close fridge | Phone on base | Umbrella out | Frame off hanger | Wine at rack | Water plants | Mean S.R. & Var | Infer. speed |
|---|---|---|---|---|---|---|---|---|---|---|---|---|
| ManipLLM [48] | 0.50 | 0.80 | 0.40 | 0.20 | 0.80 | 0.35 | 0.10 | 0.25 | 0.15 | 0.20 | 0.38 ±0.04 | 2.2 Hz |
| OpenVLA [7] | 0.65 | 0.40 | 0.75 | 0.50 | 0.80 | 0.20 | 0.35 | 0.15 | 0.10 | 0.10 | 0.40 ±0.04 | 6.3 Hz |
| $\pi_0$ [23] | 0.90 | 0.80 | **0.95** | 0.30 | 0.85 | 0.30 | 0.30 | **0.70** | 0.10 | **0.30** | 0.55 ±0.03 | 13.8 Hz |
| CogACT [22] | 0.90 | 0.80 | **0.95** | 0.50 | 0.85 | **0.50** | **0.55** | 0.45 | 0.30 | 0.25 | 0.61 ±0.04 | 9.8 Hz |
| FiS-VLA | **1.00** | **1.00** | **0.95** | **0.55** | **0.90** | **0.50** | 0.50 | **0.70** | **0.55** | 0.20 | **0.69** ±0.03 | 21.9 Hz |

# 4 Experiments

In Section 4.1, we compare the manipulation performance and inference speed of FiS-VLA with prior methods in simulated environments. The effectiveness of each component is evaluated in Section 4.2 and Appendix B. Section 4.3 presents both quantitative and qualitative results for FiS-VLA on real-world manipulation tasks, including dual-arm control under different robot configurations. Finally, in Section 4.4, we demonstrate the generalization capabilities of FiS-VLA by assessing its performance on previously unseen objects, backgrounds, and lighting conditions.

## 4.1 Simulation Experiment

**Simulation benchmark.** In order to fully evaluate our method, we tested on 10 various manipulation tasks in the RLBench [33] benchmark based on the CoppeliaSim simulator, including *Close box*, *Close Laptop*, *Toilet seat down*, *Sweep to dustpan*, *Close fridge*, *Phone on base*, *Take umbrella out*, *Frame off hanger*, *Wine at rack*, and *Water plants*. All the tasks were performed on a Franka Panda robot, using the front-view camera to get the input RGB image and point cloud. We collect the data by following pre-defined waypoints and utilizing the Open Motion Planning Library [72]. Building upon the frame-sampling technique employed in previous studies [73, 5, 28], we construct a training dataset where each task contains 100 trajectories.

**Training and evaluation details.** We compare FiS-VLA against four state-of-the-art (SOTA) VLA models, including ManipLLM [48], OpenVLA [7], $\pi_0$[23], and CogACT[22], where the latter two are dual-system methods but operate with synchronous frequencies. For baselines, we load the official pretrained parameters provided by each method and adhere to their respective fine-tuning settings. For FiS-VLA's input, the single-view RGB image is resized to $224 \times 224$, the point cloud is downsampled to 1024 points, the text instruction is derived from simulation, and the robot state is aligned with the predicted actions. FiS-VLA model is trained for 300 epochs using the AdamW optimizer [74] on 8 NVIDIA A800 GPUs, with mixed-precision training employed. All methods utilized the officially provided pre-trained parameters and underwent full fine-tuning. Following [22, 5], we evaluate all methods using 20 rollouts from the latest epoch checkpoint, repeating the evaluation three times for each task and reporting the average success rate along with the variance.

**Quantitative results.** As shown in Table 1, FiS-VLA achieves an average success rate of 69% across 10 diverse tasks, surpassing the previous SOTA methods CogACT and $\pi_0$ by margins of 8% and 14%, respectively. In particular, FiS-VLA achieves superior performance on 8 out of 10 tasks, highlighting the robustness of its action generation capabilities. By embedding the System 1 execution module within the intact VLM (System 2), FiS-VLA leverages the VLM's pretrained knowledge for action generation and enables more effective interpretation of System 2's latent feature guidance. In terms of control frequency, FiS-VLA operates at 21.9 Hz, over 2× faster than CogACT (9.8 Hz) and more than 1.6× faster than $\pi_0$ (13.8 Hz), with action chunk set to one. Note that $\pi_0$ employs a 2.6B-parameter LLM, while both CogACT and our FiS-VLA are based on a 7B-parameter LLM. The results demonstrate that our asynchronous input frequency design significantly improves the inference speed of VLA models. Moreover, the Fast-in-Slow framework facilitates effective coordination between the two systems, leading to enhanced manipulation accuracy.

## 4.2 Ablation Study

To analyze the impact of each component on overall performance within the FiS-VLA, we conduct ablation experiments on 10 RLBench tasks using the same settings as the simulation experiments.

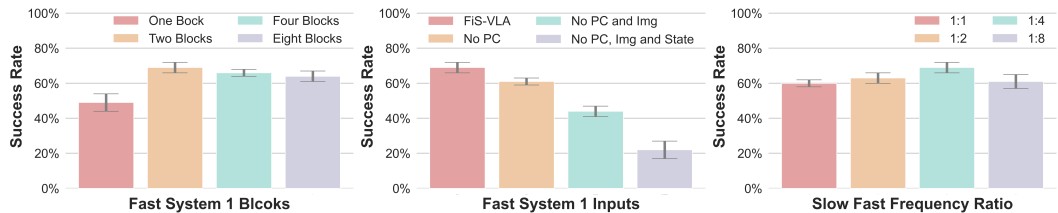

Figure 3: **Ablation study.** We investigate the impact of (1) the parameters of System 1's shared blocks within System 2, (2) different modality inputs to System 1, and (3) the operating frequency ratio between the two systems on final manipulation success rates.

**(1) The parameters of System 1's shared blocks within System 2.** In this exploration, we set the operating frequency ratio between the two systems to 1:4 and use all modality inputs. By gradually increasing the number of shared transformer blocks reconstructed from the VLM-based System 2 into the fast System 1 (from 1 to 8), we observe an improvement in manipulation performance, which tends to saturate when two blocks are used. These results show that embedding System 1 within the VLM-based System 2 enables it to inherit rich pretrained knowledge, achieving stable manipulation with relatively few parameters while maintaining high inference speed. **(2) Different modality inputs to System 1.** We compare the combinations of input information into fast System 1, evaluating the cases of using only latent features from slow System 2, adding robot state, and further incorporating 2D images and 3D point clouds. System 1 is composed of 2 transformer blocks, and the asynchronous frequency ratio between the two systems is set to 1:4. The results show that each modality substantially contributes to improving manipulation performance. The robot state provides access to the robot's internal status, while 3D point clouds enhance the understanding of geometric structure and spatial relationships. **(3) The operating frequency ratio between the two systems.** We empirically set different asynchronous frequency ratios between System 2 and System 1 (from 1:1 to 1:8). Note that in this experiment, we use 2 transformer blocks for System 1 while retaining all modality inputs. The results show that when the ratio is 1:4, FiS-VLA excels the best performance, striking a perfect balance between slow reasoning and fast action generation. This validates that the asynchronous coordination frequency design not only improves the action generation rate but also increases the informational richness of observations provided to the execution module. **(4) Training strategy.** If $\mathcal{L}_{\text{slow}}$ is removed during training, manipulation performance drops from 69% to 62%. This result underscores the importance of our dual-aware co-training strategy, which preserves the integrity and inherent reasoning capabilities of the System 2 model, thereby providing more effective latent guidance for System 1 execution. To further assess whether preserving System 2's reasoning improves FiS-VLA's action generation, we replaced discrete action supervision with task plans automatically generated by Gemini [75] and manually verified. FiS-VLA achieves a 73% average success rate with plan-based co-training, outperforming the 69% obtained using discrete actions. This suggests that explicit reasoning supervision leads to more accurate conditioning of System 1 and improved performance. More ablation experiments can be found in Appendix B.

### 4.3 Real-World Experiment

**Self-collected data.** For dual-arm tasks, we evaluate four tasks on the Agilex Robot and AlphaBot respectively, each equipped with three camera views: a static exterior view, a right-wrist view, and a left-wrist view. On the Agilex Robot, we conduct the following four tasks: 1) *Pick objects and place in basket*, 2) *Lift ball and place in basket*, 3) *Place bottles at rack*, 4) *Wipe blackboard*. On the AlphaBot, we perform another set of four tasks: 1) *Pick bowl and place object*, 2) *Handover object and place*, 3) *Pour water and move cup*, 4) *Fold towel and place in bucket*. For each task, we collect 100 demonstrations via master-puppet teleoperation, with objects placed in varying positions on the table to ensure diversity. Additional implementation details can be found in the Appendix A.

**Training and evaluation details.** We evaluate FiS-VLA against $\pi_0$ [23], using the same training setup as in simulation, with the exception of three-view RGB inputs for real-world dual-arm tasks. Evaluation is conducted using the final checkpoint over 20 rollouts across varied tabletop positions. Note that we control the Agilex Robot using end-effector poses and the AlphaBot using joint positions, demonstrating our model's effectiveness across different robot control paradigms.

Table 2: **Comparison of FiS-VLA and $\pi_0$ in real-world scenarios.** We train all methods in a single-task setting [26] and report the success rates. Success is determined by human evaluation based on whether the task is completed.

| Models | Agilex Dual-Arm Robot Task | | | | | AlphaBot Dual-Arm Robot Task | | | | |
| --- | --- | --- | --- | --- | --- | --- | --- | --- | --- | --- |
| | Pick and place | Lift ball and place | Place bottles at rack | Wipe blackboard | Mean S.R. ↑ | Pick bowl and place object | Handover and place | Pour water and move | Fold towel and place | Mean S.R. ↑ |
| $\pi_0$ [23] | 0.70 | **0.75** | 0.55 | 0.35 | 0.59 | 0.65 | 0.75 | 0.65 | 0.40 | 0.61 |
| FiS-VLA | **0.80** | **0.75** | **0.70** | **0.45** | **0.68** | **0.80** | **0.80** | **0.75** | **0.60** | **0.74** |

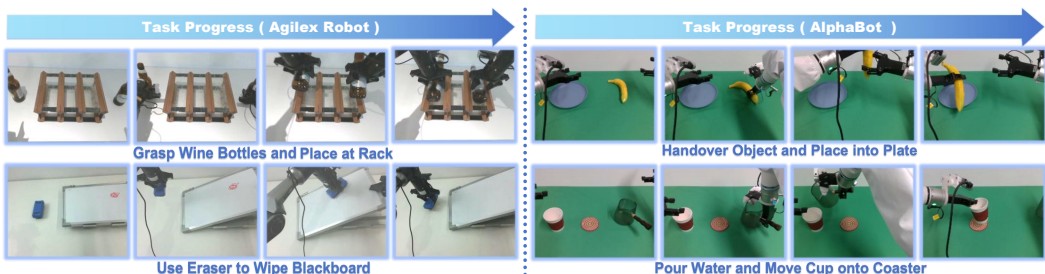

Figure 4: Visualization of real-world experiments with Agilex and AlphaBot dual-arm robots.

**Quantitative and qualitative results.** As shown in Table 2, FiS-VLA consistently outperforms the baseline $\pi_0$ across eight real-world tasks. On the Agilex Robot, FiS-VLA achieves a mean success rate of 68%, compared to 59% for $\pi_0$. Notably, FiS-VLA achieves significantly higher success rates in complex manipulation tasks requiring precise spatial reasoning. For example, in the *Place Bottles at Rack* task, our method attains a 70% success rate compared to $\pi_0$ 55%. Similarly, on the AlphaBot platform, FiS-VLA achieves a higher mean success rate of 74%, surpassing $\pi_0$ 61%. The greatest improvement is seen in the *Fold towel and put* task, which involves manipulating deformable objects. Qualitative results in Figure 4 showcase FiS-VLA's ability to execute diverse tasks across robots, including sequential bottle manipulation and blackboard erasing on Agilex, as well as fine-grained actions like pouring water on AlphaBot. These outcomes highlight the model's effective coordination of high-level reasoning and low-latency control, enabling adaptive behavior in real-world settings. Additional visualizations and failure cases are provided in Appendix C and D, respectively.

## 4.4 Generalization Experiment

To evaluate the generalization of FiS-VLA in real-world settings, we conduct three test scenarios involving unseen manipulated objects, complex backgrounds, and varying lighting conditions, as shown in Table 3 and Figure 5. We compare FiS-VLA with the baseline model $\pi_0$ on two tasks: *place bottles at rack* using the Agilex platform and *Pick bowl and place object* using the AlphaBot platform. **(1) Unseen manipulated objects.** This experiment evaluates the generalization of FiS-VLA to novel object instances. For example, the banana is replaced with a visually distinct hot dog bun. FiS-VLA demonstrates a smaller performance drop compared to $\pi_0$ across both platforms. Notably, on AlphaBot, FiS-VLA experiences only a 19% reduction in accuracy, whereas $\pi_0$ suffers a 38% drop. These results demonstrate that under the proposed FiS-VLA dual-system paradigm, embedding the System 1 execution module within the VLM-based System 2 allows it to better inherit the rich pretrained knowledge of the VLM and more effectively interpret the high-level reasoning latent features provided by System 2. **(2) Complex backgrounds.** To simulate distracting environments, we introduce visually cluttered scenes containing irrelevant objects such as mugs, hamburgers, and bottles. These test whether the model can comprehend human instructions and task-relevant information while ignoring distractions. FiS-VLA demonstrates more stable performance than $\pi_0$, with only a 25% drop in accuracy on AlphaBot and a 29% drop on Agilex. This validates that System 2 of FiS-VLA excels at focusing on semantically relevant objects through contextual reasoning, while System 1 ensures execution remains aligned with real-time visual cues. **(3) Varying lighting conditions.** Lighting variation is a common real-world challenge that often negatively impacts the model's perception. In this setting, FiS-VLA still demonstrates strong generalization capabilities, achieving over 50% manipulation success on both robotic platforms. These results highlight the

Table 3: **Generalization experiments.** "Object", "Background", and "Lighting" refer to unseen manipulated objects, complex backgrounds, and illumination disruption, respectively.

| Task | Place Bottles at Rack | | Pick Bowl and Place Object | |
|---|---|---|---|---|
| Robot | Agilex Robot | | AlphaBot | |
| Models | FiS-VLA | $\pi_0$ [23] | FiS-VLA | $\pi_0$ [23] |
| Original | 0.70 | 0.55 | 0.80 | 0.65 |
| Object | 0.55 (-21%) | 0.40 (-27%) | 0.65 (-19%) | 0.40 (-38%) |
| Background | 0.50 (-29%) | 0.35 (-36%) | 0.60 (-25%) | 0.40 (-38%) |
| Lighting | 0.50 (-29%) | 0.40 (-27%) | 0.55 (-31%) | 0.35 (-46%) |

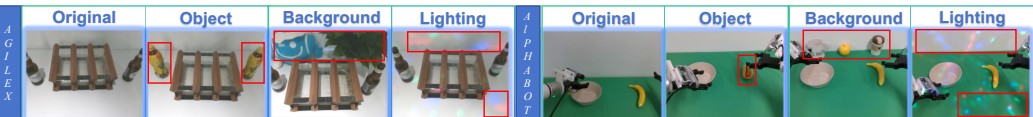

Figure 5: Visualization of generalization setting with key differences highlighted using red box.

importance of the heterogeneous modality input design in FiS-VLA's dual systems, which enhances robustness to perceptual perturbations.

## 5 Conclusion and Limitation

In this paper, we introduce Fast-in-Slow, a novel dual-system VLA foundation model that embeds a fast execution module (System 1) seamlessly within a VLM-Based slow reasoning model (System 2), thereby achieving high-frequency action generation while maintaining the reasoning capability of pre-trained VLMs. We conducted a comprehensive investigation into the dual-system architecture, analyzing their divergent task objectives, asynchronous operating frequencies, and heterogeneous input modalities. Furthermore, we propose a novel dual-aware co-training strategy that enables joint optimization of both systems. However, FiS-VLA statically configures the shared parameters of System 1 within System 2 and the collaboration frequency between the two systems. We hypothesize that enabling dynamic adaptation of these factors based on task demands and environmental complexity could lead to a more robust and generalizable model, which will be a key focus of our future work. Finally, the social impact of our work is detailed in Appendix E.

## Acknowledgement

This work was supported by the National Natural Science Foundation of China (62476011), and in part by the Research Grants Council of the Hong Kong Special Administrative Region, China, under Project AoE/E-407/24-N, as well as the InnoHK Clusters via Hong Kong Center for Logistics Robotics.

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

**Appendix A Additional Dataset Details.** In this section, the construction of real-world datasets for large-scale pretraining is described. Subsequently, the self-collected simulator and real-world datasets used for downstream task fine-tuning are introduced.

**Appendix B Additional Quantitative Results.** In this section, we present additional ablation studies, which include an investigation into the impact of action chunking on manipulation performance and control frequency, as well as a deeper exploration of heterogeneous modality inputs for System 1 and System 2. Moreover, we conducted experiments to validate the effectiveness of both the shared parameters in FiS-VLA and the proposed method using a small-scale LLM backbone. Finally, we report the detailed success rates for each task across all ablation experiments, including those presented in both the main paper and the appendix.

**Appendix C Additional Qualitative Results.** In this section, we provide additional visualizations of both simulation and real-world tasks. Compared to the main paper, this part offers a more detailed illustration of the execution process for each task.

**Appendix D Failure Case Analysis.** In this section, we analyze failure cases observed when deploying FiS-VLA to control dual-arm robots in real-world scenarios.

**Appendix E Broader Impact.** A brief discussion on the potential broader impact of our work.

# A    Additional Dataset Details

## A.1    Large-scale pretraining dataset

Similar to RDT [29] and CogACT [22], we assemble a large-scale pre-training dataset by integrating existing open-source robotic datasets. Our pre-training corpus consists of 37 datasets, totaling 860k trajectories and 36 million frames. By including both single-arm and recent dual-arm datasets such as RDT and RoboMIND [32], our pre-training corpus enhances the model's ability to generalize across diverse robotic control configurations. Table 4 provides a comprehensive list of all datasets used in pre-training along with their corresponding sampling weights. Both the number of trajectories and sampling weights can be automatically adjusted during dataset assembly. Following the preprocessing pipeline introduced in [7], we reformulate the dataset to preserve both end-effector trajectory control and joint position control for robot actions. Regarding observations, due to structural discrepancies across datasets, we use only single-view 2D RGB images as visual inputs during pre-training. During fine-tuning, FiS-VLA supports both single-view and multi-view inputs, depending on the task requirements and robot hardware configuration. For instance, in AgileX and AlphaBot dual-arm robot tasks, we use three camera views: one exterior camera and two wrist cameras, in order to mitigate occlusions caused by the robot arms. Furthermore, leveraging our heterogeneous modality design, the Fast System 1 of FiS-VLA is equipped to process point cloud data derived from exterior-view depth maps, computed using the camera's intrinsic and extrinsic parameters. It is worth noting that, although the number and modality of input images differ between pre-training and fine-tuning, the training objectives and overall training recipe remain consistent. Consequently, this variation does not degrade downstream manipulation performance; instead, the integration of multi-view and multimodal inputs contributes to a more robust manipulation policy.

## A.2    Simulation dataset

We follow the simulation setup used in PerAct and RVT, employing CoppeliaSim to collect 10 RLBench[33] tabletop tasks, which are executed using a Franka Panda robot equipped with a two-finger parallel gripper. These tasks cover pick-and-place, tool use, articulated object manipulation, and several precise control tasks, including: *Close box*, *Close laptop*, *Toilet seat down*, *Sweep to dustpan*, *Close fridge*, *Phone on base*, *Take umbrella out*, *Frame off hanger*, *Wine at rack*, and *Water plants*, similar to prior work [30, 28]. Although the simulator environment includes multiple RGB-D cameras, we only leverage the front-view camera to obtain RGB images and point cloud inputs. Following previous work [5, 73], we collect 100 trajectories per task using pre-defined waypoints and the Open Motion Planning Library, and apply the same frame-sampling method to extract keyframes for building the training dataset. The visualizations of the execution process in simulation are shown in Figure 8 and Figure 9.

Table 4: **The dataset name and sampling weight used in our mixed large-scale pretraining dataset.**

| Training Dataset Mixture | |
|---|---|
| Fractal [40] | 6.8% |
| Kuka [76] | 10.5% |
| Bridge[77, 78] | 4.9% |
| Taco Play [79, 80] | 2.5% |
| Jaco Play [81] | 0.4% |
| Berkeley Cable Routing [82] | 0.2% |
| Roboturk [83] | 2.0% |
| Viola [84] | 0.8% |
| Berkeley Autolab UR5 [85] | 1.0% |
| Toto [86] | 1.7% |
| Language Table [87] | 3.7% |
| Stanford Hydra Dataset [88] | 3.8% |
| Austin Buds Dataset [89] | 1.8% |
| NYU Franka Play Dataset [90] | 0.7% |
| Furniture Bench Dataset [91] | 2.1% |
| UCSD Kitchen Dataset [92] | <0.1% |
| Austin Sailor Dataset [93] | 1.9% |
| Austin Sirius Dataset [94] | 1.5% |
| DLR EDAN Shared Control [95] | <0.1% |
| IAMLab CMU Pickup Insert [96] | 0.7% |
| UTAustin Mutex [97] | 1.9% |
| Berkeley Fanuc Manipulation [98] | 0.6% |
| CMU Stretch [99] | 0.1% |
| BC-Z [100] | 6.3% |
| FMB Dataset [101] | 6.0% |
| DobbE [102] | 1.2% |
| DROID [20] | 14.2% |
| Stanford Kuka Dataset [103] | 0.3% |
| Stanford Robocook Dataset [104] | 0.2% |
| Columbia Cairlab Pusht Real [3] | <0.1% |
| UCSD Pick and Place | 0.8% |
| Maniskill [105] | 7.5% |
| Berkeley RPT [106] | <0.1% |
| QUT Dexterous Manipulation [107] | <0.1% |
| RoboSet [108] | 5.2% |
| BridgeData V2 [78] | 9.3% |
| RoboMind [32] | 1.2% |

## A.3  Self-collected real-world dataset

For real-world experiments, we evaluate four tasks each on the Agilex Robot and the AlphaBot robot. Below, we detail the hardware configurations, data collection protocols, and task setting for both platforms.

**Agilex robot setup.** As summarized in Table 5, the Agilex Robot is equipped with two 6-DoF arms mounted on a mobile base. As shown in Figure 6, two Orbbec DABAI cameras capture the left and right wrist views, while a RealSense 435 camera mounted overhead provides exterior-view RGB images and point cloud data. All cameras record at 30 Hz. For trajectory recording and control, we use end-effector poses. The four tasks conducted on the Agilex Robot are as follows:

1) *Pick objects and place in basket*. The robot uses both arms to pick up two objects according to a language command and place them into a container. This task assesses the model's understanding of spatial positioning.

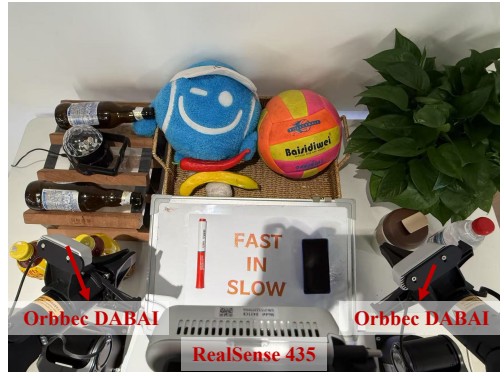 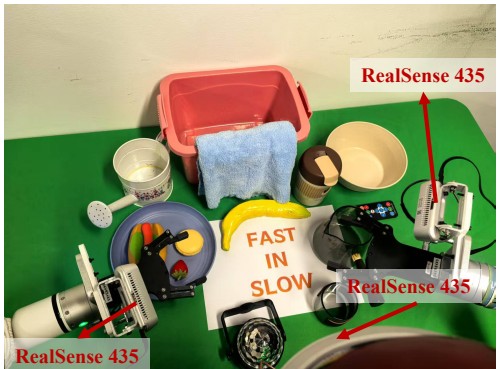

(a) Agilex dual-arm robot          (b) AlphaBot dual-arm robot

Figure 6: **Real-world assets and camera configurations.** We present visualizations of the real-world assets and camera setups used in the Agilex and AlphaBot dual-arm robot tasks, respectively.

Table 5: **The hardware setups of the two dual-arm robots, including the number of joints and their corresponding angle ranges of motion.**

| Agilex dual-arm robot | | AlphaBot dual-arm robot | |
|---|---|---|---|
| Joint number | Angle range | Joint number | Angle Range |
| J1 | $-154° \sim +154°$ | J1 | $-178° \sim +178°$ |
| J2 | $0° \sim +195°$ | J2 | $-130° \sim +130°$ |
| J3 | $-175° \sim 0°$ | J3 | $-178° \sim +178°$ |
| J4 | $-106° \sim +106°$ | J4 | $-135° \sim +135°$ |
| J5 | $-75° \sim +75°$ | J5 | $-178° \sim +178°$ |
| J6 | $-100° \sim +100°$ | J6 | $-128° \sim +128°$ |
| - | - | J7 | $-180° \sim +180°$ |

2) *Lift ball and place in basket.* The robot must synchronize both arms to grasp a ball held between the grippers and transport it without slippage. This task evaluates dual-arm coordination.

3) *Place bottles at rack.* Each arm grasps a bottle from its side, rotates it, and aligns it parallel to the rack. This task tests inter-object relationship reasoning and precise rotational manipulation.

4) *Wipe blackboard.* One arm holds the board while the other erases red marker using an eraser. This setup tests precise, coordinated actions in dual-arm scenarios.

**AlphaBot robot setup.** As shown in Table 5, the AlphaBot leverages two 7-DoF arms mounted on a mobile base. As shown in Figure 6, three RealSense 435 cameras are used to capture the left wrist, right wrist, and exterior views, while only the exterior view is used for point cloud generation. All modalities are recorded at 30 Hz. To evaluate the model's robustness to different control schemes, we adopt joint position control for both trajectory collection and inference execution. For each task, we collect 100 demonstrations using master-puppet teleoperation, with object positions randomized on the table to promote data diversity. Language instructions are manually created and diversified via augmentation. The four tasks evaluated on the AlphaBot include:

1) *Pick bowl and place object.* The robot uses its left arm to pick up a bowl and its right arm to pick up an object, placing the object into the bowl. This task involves coordinated dual-arm manipulation, where each arm performs distinct, asymmetric roles.

2) *Handover object and place.* The right arm picks up an object and hands it to the left arm. The arms must avoid collisions and ensure proper grasp alignment. The left arm then places the object into a plate. This task serves as a comprehensive benchmark for evaluating the model's capabilities in 3D perception, grasp reasoning, and dual-arm motion planning. It poses significant challenges while remaining highly practical for real-world bimanual manipulation scenarios.

3) *Pour water and move cup.* The robot grasps a cup handle with its right arm, rotates it to pour water into another cup, then moves the receiving cup to a coaster. This task combines high-precision

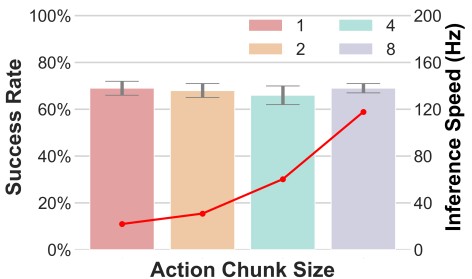 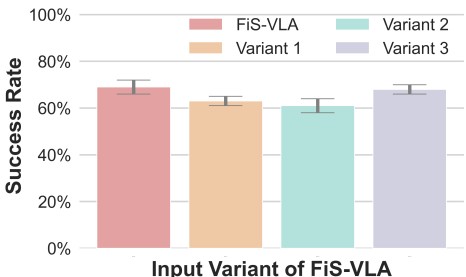

Figure 7: **Ablation studies on action chunk size and input variants of FiS-VLA.** (Left) Impact of different action chunk sizes on success rate and inference speed. While increasing action chunk size leads to improved inference speed, success rate remains relatively stable. (Right) Comparison of success rates among FiS-VLA and its input variants, showing FiS-VLA achieves the best performance.

pose control, physical reasoning, and multi-stage planning, making it a representative benchmark for evaluating precision-oriented manipulation capabilities

4) *Fold towel and place in bucket*. The robot folds a deformable towel using both arms, then places it into a bucket. This task evaluates coordinated manipulation of deformable objects.

# B    Additional Quantitative Results

## B.1    Action chunking for robust and high-Frequency robot control

In closed-loop control of robots, a key challenge is the compounding of errors, where an early mistake can cascade into subsequent decisions, ultimately driving the system's observations far from the training distribution and leading to unrecoverable failures [109]. To mitigate this issue, inspired by the concept of *action chunking*, researchers have explored predicting multiple actions at once [29, 22, 23, 110]. This approach reduces the number of decision points along a trajectory, thereby decreasing the opportunity for error accumulation. Moreover, it enables higher effective control frequencies, resulting in smoother and more continuous robot motions. By considering sequences of actions jointly, the model can enforce temporal consistency and avoid abrupt changes that could physically damage the robot. In this work, we investigate the effect of action chunking by predicting future action sequences of length $H$ ranging from one to eight, as illustrated in Figure 7 and Table 11. We observe that the performance of FiS-VLA remains stable across different values of $H$, while the control frequency increases proportionally. Notably, when predicting eight future actions in a single step, the theoretical control frequency reaches up to **117.7 Hz**, demonstrating the potential of our method for high-speed, high-fidelity robotic control.

## B.2    Multi-modal input configuration analysis

In our study, we found that incorporating multi-modal inputs consisting of 2D images, 3D point clouds, and robot state information into System 1 of FiS-VLA significantly improves execution accuracy. Based on this observation, we conducted a more comprehensive investigation to evaluate the impact of different combinations of these modalities when provided to System 1 and System 2. We refer to these configurations as the **input variants of FiS-VLA**. The results are presented in Figure 7 and Table 12. In **Variant 1**, System 2 receives language instructions, 2D images, and 3D point clouds, while System 1 takes 2D images and robot state as input. This configuration leads to slightly lower control accuracy compared to the original FiS-VLA. In **Variant 2**, the robot state input is moved from System 1 to System 2, resulting in a marginal performance drop relative to Variant 1. Finally, in **Variant 3**, we explored a configuration where both System 1 and System 2 receive 2D images and 3D point clouds. Additionally, System 1 receives the robot state and System 2 receives the language instruction. This setup achieves performance that is nearly equivalent to the original FiS-VLA. These results demonstrate the robustness and flexibility of the FiS-VLA architecture in integrating multi-modal information for high-precision robotic control.

## B.3 Effectiveness of Parameter Sharing

To evaluate the effectiveness of our parameter-sharing design, we conducted the following ablation studies. **Experiment 1**: The VLM serves as System 2, and the last two LLM layers are duplicated to form an independent System 1. This configuration supports both autoregressive generation from System 2's VLM and diffusion-based action prediction through the duplicated LLM layers. The entire system is trained using identical settings with FiS-VLA. **Experiment 2**: This experiment mirrors the setup of Experiment 1, with the difference being that four LLM layers are duplicated instead of two. As shown in the Table 6, both variants underperform compared to FiS-VLA. We attribute this to feature misalignment. The LLM was pretrained across all 32 layers, enabling hierarchical (layer-by-layer) processing. Duplicating the last few layers breaks this chain (e.g., in Experiment 1, the new 31st layer in System 1 must process outputs from System 2's 32nd layer), despite never being pretrained to do so. This disrupts feature compatibility and limits performance. In contrast, FiS-VLA's parameter-sharing keeps the LLM's original structure and information flow intact. System 1 remains embedded within System 2, preserving pretrained knowledge while enabling action execution. This design proves more efficient and effective than separated architectures.

Table 6: **Results of additional Transformer blocks as System 1 on RLBench.**

|  | Close box | Close laptop lid | Toilet seat down | Sweep to dustpan | Close fridge | Phone on base | Umbrella out | Frame off hanger | Wine at rack | Water plants | Mean S.R. & Var |
|---|---|---|---|---|---|---|---|---|---|---|---|
| Experiment 1 | 0.80 | 0.95 | **0.95** | **0.55** | 0.80 | **0.55** | 0.40 | **0.70** | 0.30 | 0.10 | 0.61 ±0.03 |
| Experiment 2 | 0.90 | 0.85 | 0.90 | 0.35 | **0.90** | 0.45 | 0.45 | 0.50 | 0.45 | 0.10 | 0.59 ±0.05 |
| FiS-VLA | **1.00** | **1.00** | **0.95** | **0.55** | **0.90** | 0.50 | **0.50** | **0.70** | **0.55** | **0.20** | **0.69** ±0.03 |

## B.4 Performance of FiS-VLA with a Smaller-Scale LLM

We further conducted an additional comparison using the 2.7B Phi-2 model as the LLM backbone. To ensure fairness, FiS-VLA (2.7B) was also pretrained on the same assembled robotic datasets under identical settings. As shown in the Table below, the compact-sized FiS-VLA also achieves satisfactory performance, which demonstrates the effectiveness of our approach and its generalizability to different VLM backbones.

Table 7: **Results of FiS-VLA with 2.7B and 7B LLMs on RLBench.**

|  | Close box | Close laptop lid | Toilet seat down | Sweep to dustpan | Close fridge | Phone on base | Umbrella out | Frame off hanger | Wine at rack | Water plants | Mean S.R. & Var |
|---|---|---|---|---|---|---|---|---|---|---|---|
| FiS-VLA (2.7B) | 0.90 | 0.85 | 0.90 | **0.55** | 0.80 | **0.55** | 0.40 | 0.65 | 0.45 | 0.15 | 0.62 ±0.03 |
| FiS-VLA (7B) | **1.00** | **1.00** | **0.95** | **0.55** | **0.90** | 0.50 | **0.50** | **0.70** | **0.55** | **0.20** | **0.69** ±0.03 |

## B.5 The detailed results for each experimental setting

We have presented all the results of the ablation studies in both the main paper and the appendix in a fine-grained manner, as shown in Table 8 to Table 12.

Table 8: **Results of different fast System 1 blocks on RLBench.** The results in this table correspond to the first subplot of Figure 3 in the main paper.

| Fast System 1 blocks | Close box | Close laptop lid | Toilet seat down | Sweep to dustpan | Close fridge | Phone on base | Umbrella out | Frame off hanger | Wine at rack | Water plants | Mean S.R. & Var |
|---|---|---|---|---|---|---|---|---|---|---|---|
| One block | 0.70 | 0.55 | 0.95 | 0.55 | 0.80 | 0.05 | 0.20 | 0.70 | 0.30 | 0.10 | 0.49 ±0.05 |
| Two blocks (FiS-VLA) | **1.00** | **1.00** | 0.95 | **0.55** | 0.90 | **0.50** | 0.50 | 0.70 | **0.55** | 0.20 | **0.69** ±0.03 |
| Four blocks | **1.00** | 0.90 | **1.00** | 0.45 | 0.85 | 0.35 | **0.60** | **0.75** | 0.40 | 0.25 | 0.66 ±0.02 |
| Eight blocks | 0.90 | 0.80 | 0.95 | **0.55** | **0.95** | 0.45 | 0.45 | 0.65 | 0.40 | **0.30** | 0.64 ±0.03 |

Table 9: **Results of different fast System 1 input on RLBench.** The results in this table correspond to the second subplot of Figure 3 in the main paper.

| Fast System 1 input | Close box | Close laptop lid | Toilet seat down | Sweep to dustpan | Close fridge | Phone on base | Umbrella out | Frame off hanger | Wine at rack | Water plants | Mean S.R. & Var |
|---|---|---|---|---|---|---|---|---|---|---|---|
| FiS-VLA | **1.00** | **1.00** | 0.95 | **0.55** | **0.90** | **0.50** | 0.50 | 0.70 | **0.55** | **0.20** | **0.69** ±0.03 |
| No PC | 0.90 | 0.90 | 0.85 | 0.35 | 0.85 | 0.20 | **0.55** | **0.80** | 0.50 | 0.15 | 0.61 ±0.02 |
| No PC and Img | 0.45 | 0.45 | **0.95** | 0.30 | 0.75 | 0.10 | 0.50 | 0.50 | 0.30 | 0.10 | 0.44 ±0.03 |
| No PC, Img and State | 0.50 | 0.30 | 0.15 | 0.00 | 0.65 | 0.05 | **0.55** | 0.45 | 0.00 | 0.00 | 0.22 ±0.05 |

Table 10: **Results of different slow fast frequency ratio on RLBench.** The results in this table correspond to the third subplot of Figure 3 in the main paper.

| Frequency ratio | Close box | Close laptop lid | Toilet seat down | Sweep to dustpan | Close fridge | Phone on base | Umbrella out | Frame off hanger | Wine at rack | Water plants | Mean S.R. & Var |
|---|---|---|---|---|---|---|---|---|---|---|---|
| 1:1 | 0.95 | 0.80 | 0.85 | 0.30 | **1.00** | 0.40 | 0.40 | 0.65 | 0.45 | 0.20 | 0.60 ±0.02 |
| 1:2 | 0.90 | 0.85 | **1.00** | 0.30 | 0.90 | 0.30 | **0.55** | 0.70 | 0.45 | **0.30** | 0.63 ±0.03 |
| 1:4 (FiS-VLA) | **1.00** | **1.00** | 0.95 | **0.55** | 0.90 | **0.50** | 0.50 | 0.70 | **0.55** | 0.20 | **0.69** ±0.03 |
| 1:8 | 0.85 | 0.90 | 0.95 | **0.55** | 0.95 | 0.30 | 0.45 | **0.85** | 0.15 | 0.10 | 0.61 ±0.04 |

Table 11: **Results of different action chunk size on RLBench.** The results in this table correspond to the first subplot of Figure 7 in the appendix.

| Action chunk size | Close box | Close laptop lid | Toilet seat down | Sweep to dustpan | Close fridge | Phone on base | Umbrella out | Frame off hanger | Wine at rack | Water plants | Mean S.R. & Var |
|---|---|---|---|---|---|---|---|---|---|---|---|
| 1 | **1.00** | **1.00** | 0.95 | **0.55** | **0.90** | 0.50 | 0.50 | **0.70** | **0.55** | 0.20 | **0.69** ±0.03 |
| 2 | **1.00** | 0.85 | **1.00** | 0.50 | 0.85 | 0.40 | **0.75** | 0.65 | 0.35 | 0.40 | 0.68 ±0.03 |
| 4 | **1.00** | 0.90 | **1.00** | 0.25 | **0.90** | 0.70 | 0.65 | 0.55 | 0.25 | 0.40 | 0.66 ±0.04 |
| 8 | 0.70 | 0.90 | 0.95 | 0.30 | **0.90** | 0.70 | 0.65 | 0.60 | 0.50 | **0.65** | **0.69** ±0.02 |

Table 12: **Results of different input variants of FiS-VLA on RLBench.** The results in this table correspond to the second subplot of Figure 7 in the appendix.

| Input variant | Close box | Close laptop lid | Toilet seat down | Sweep to dustpan | Close fridge | Phone on base | Umbrella out | Frame off hanger | Wine at rack | Water plants | Mean S.R. & Var |
|---|---|---|---|---|---|---|---|---|---|---|---|
| FiS-VLA | **1.00** | **1.00** | 0.95 | **0.55** | **0.90** | 0.50 | 0.50 | **0.70** | **0.55** | 0.20 | **0.69** ±0.03 |
| Variant 1 | 0.95 | 0.90 | **1.00** | 0.45 | 0.85 | 0.40 | **0.55** | **0.70** | 0.45 | 0.05 | 0.63 ±0.02 |
| Variant 2 | 0.90 | 0.90 | 0.95 | 0.35 | 0.80 | 0.50 | 0.45 | 0.55 | 0.45 | **0.20** | 0.61 ±0.03 |
| Variant 3 | **1.00** | 0.90 | 0.95 | 0.50 | 0.85 | **0.70** | 0.50 | 0.65 | **0.55** | **0.20** | 0.68 ±0.02 |

# C  Additional Visualizations

This section presents keyframe visualizations of FiS-VLA performing tasks in the RLBench simulator and on two real-world robotic platforms: the Agilex Robot and AlphaBot. These visualizations complement the experimental results discussed in the main paper. Figures 8 and 9 depict the execution of tasks by the Franka Panda Arm within the RLBench simulation environment. In this simulated setting, ten representative tasks are demonstrated, each broken down into key execution steps. These keyframes intuitively illustrate FiS-VLA's action selection and execution logic at various stages, showcasing its robust capabilities in sequential action prediction and gripper state control.

In real-world scenarios, FiS-VLA is evaluated across eight diverse tasks on two distinct robotic platforms. Figures 10 and 11 provide keyframe snapshots of task execution by the Agilex Robot and AlphaBot, respectively. On the Agilex Robot, tasks such as *Place bottles at rack* and *Wipe blackboard* highlight FiS-VLA's ability to perform reliable dual-arm coordination and spatial generalization in cluttered, unstructured environments. Furthermore, we evaluate FiS-VLA on long-horizon, multistage tasks, such as *Handover object and place* and *Pour water and move cup*, which require managing sequential dependencies and diverse manipulation skills. In these scenarios, FiS-VLA demonstrates consistent performance across stages and effectively utilizes dual-arm collaboration when necessary, enabling the successful execution of tasks that require both synchronized actions and long-term planning. These results collectively validate FiS-VLA's strong generalization across domains and platforms, reinforcing its promise as a versatile and scalable visuomotor policy for real-world robotic manipulation.

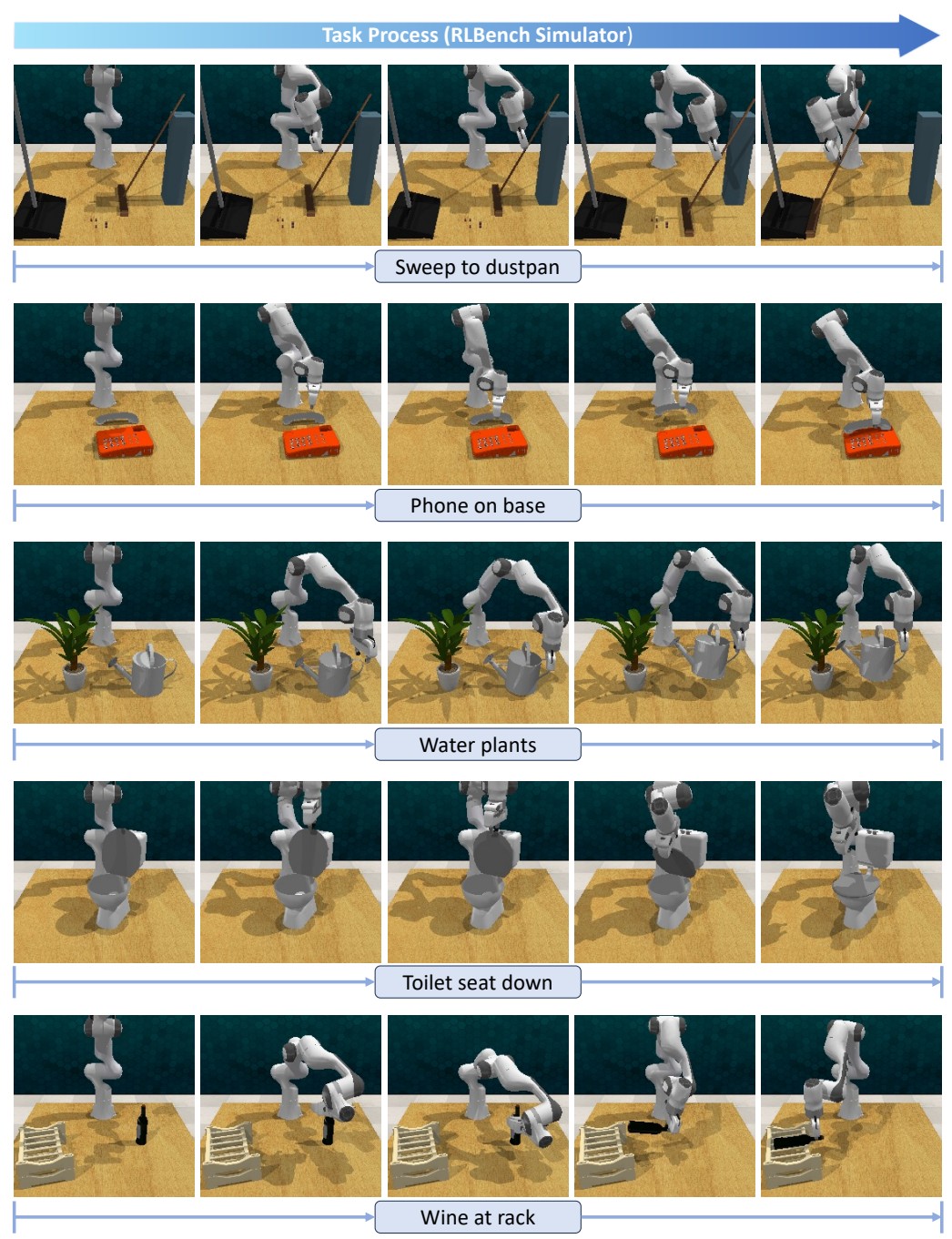

Figure 8: **RLBench visualization**. We visualize key frames of the agent's execution process from the front perspective.

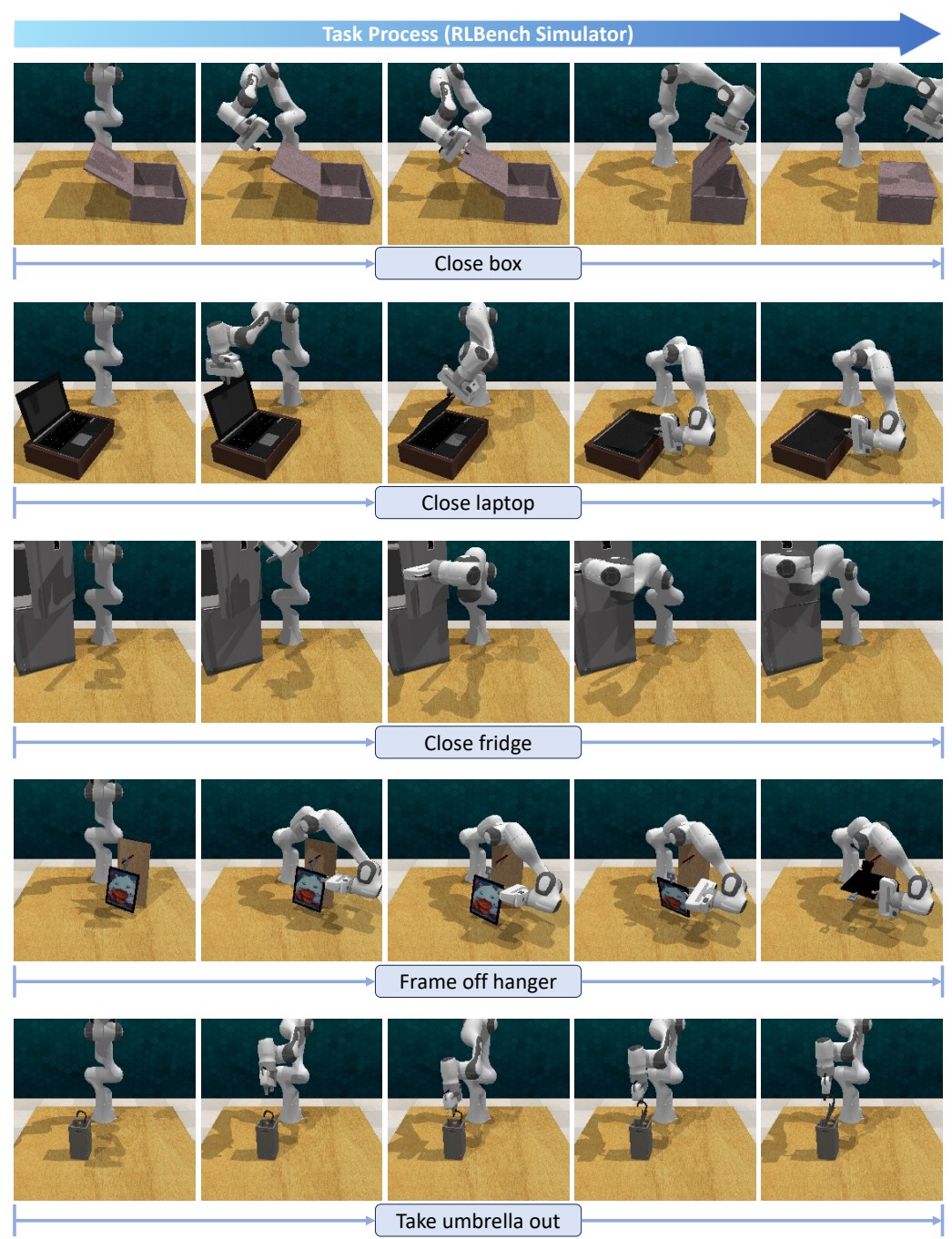

Figure 9: **RLBench visualization**. We visualize key frames of the agent's execution process from the front perspective.

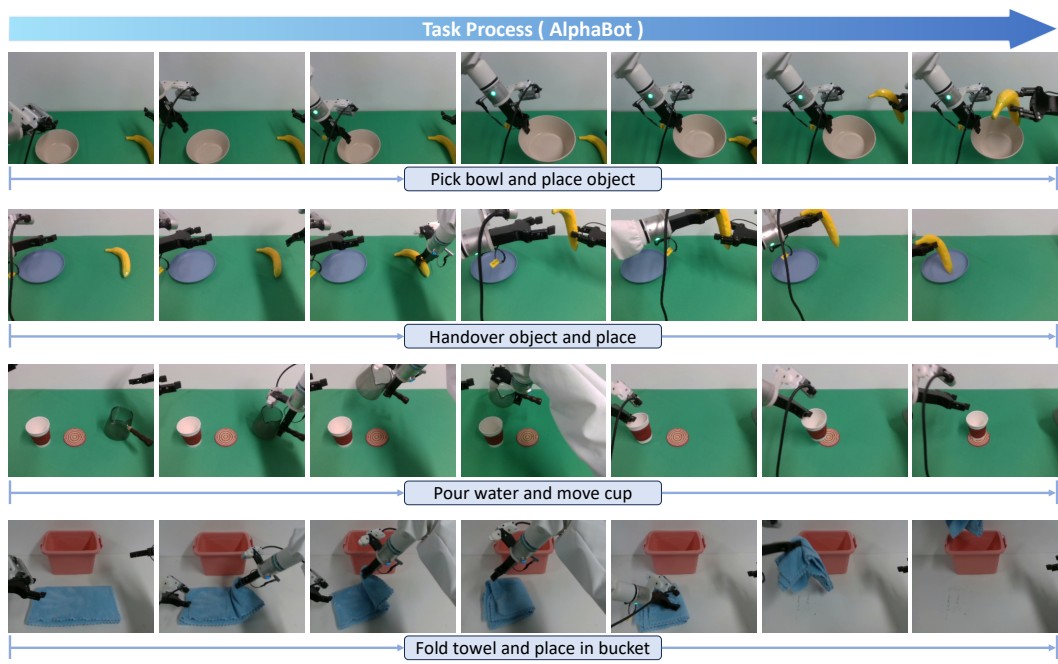

Figure 10: **Agilex robot task execution visualization**. We visualize key frames of the agent's execution process from a static exterior view.

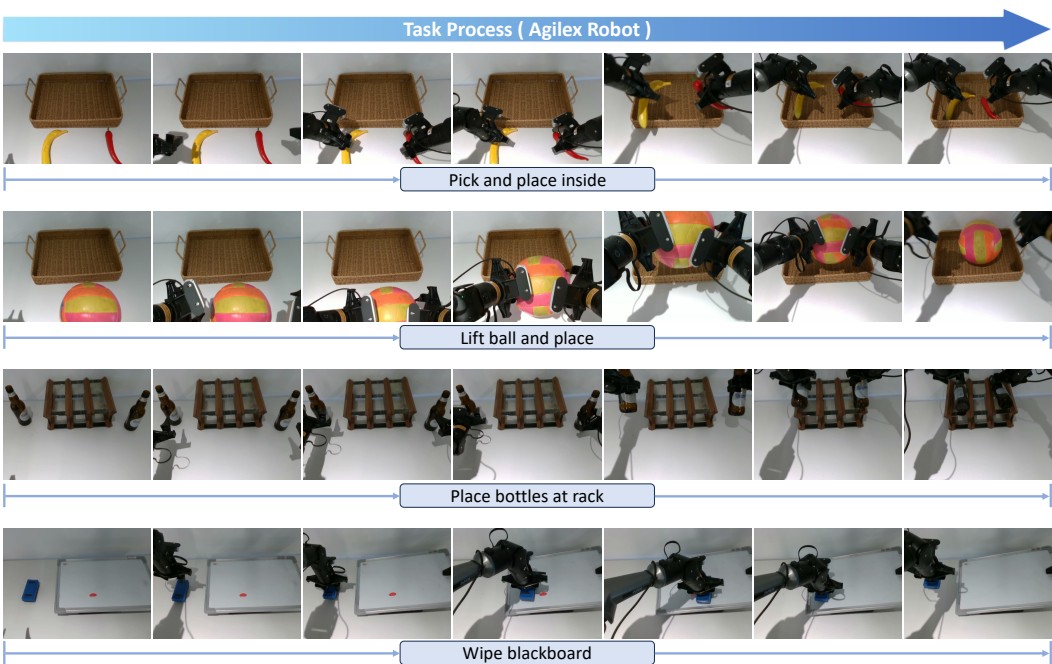

Figure 11: **AlphaBot task execution visualization**. We visualize key frames of the agent's execution process from a static exterior view.

## D    Failure Case Analysis.

Through real-world experiments on the AlphaBot platform, we observe four specific failure cases encountered by our proposed FiS-VLA, as visualized in Figure 12. Red bounding boxes highlight the critical error frames during each execution sequence.

1) The first case involves a **bimanual collision** during the *Handover object and place into plate* task. The left and right arms interfere with each other while attempting to transfer the object, indicating insufficient inter-arm motion coordination and suboptimal wrist camera placement.

2) The second case, observed in the *Fold towel and place in bucket* task, is related to an error in **manipulation height**. The predicted joint positions fail to control gripper contact with the towel, revealing the difficulty of height prediction when dealing with thin, deformable objects.

3) The third case, from the *Pick bowl and place object* task, reflects a failure in **manipulation position**. The robot mispredicts the location of the banana, resulting in a failed grasp attempt.

4) The fourth case presents a **handover rotation error** in the *Handover object and place into plate* task. The right arm rotates the object into an unsuitable orientation, preventing the left arm from executing a stable handover grasp.

These issues can be mitigated by collecting more high-quality demonstrations and incorporating efficient constraints during training to improve robustness in real-world control. Furthermore, enabling our System 2 to recognize and correct failure actions will be a key direction for future work.

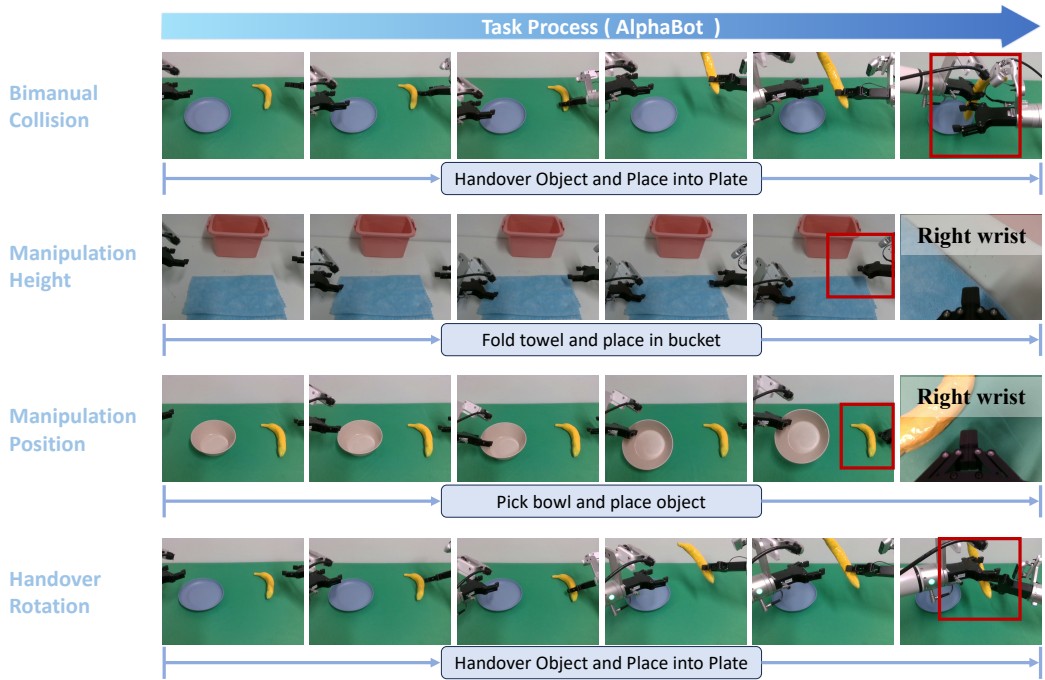

Figure 12: **Failure case visualization.** We visualize the failure cases observed in four real-world experiments, with key error frames during execution highlighted using red bounding boxes.

## E    Broader Impact

Our work proposes a foundation model for robotic manipulation that integrates high-level reasoning and low-latency action execution within a unified end-to-end Vision-Language-Action (VLA) framework. While the FiS-VLA model improves control efficiency and leverages pretrained reasoning capabilities, it may introduce potential risks when deployed in real-world environments. These risks include safety concerns in high-speed closed-loop control and unsafe behaviors resulting from the misinterpretation of human instructions. To mitigate such risks, future deployments should incorporate strict safety constraints and task-specific operational boundaries. Furthermore, our framework provides robust control and generalizable reasoning capabilities for robotic assistance in domains such as elder care and home automation, where responsiveness and reliability are critical.

