# OpenReview forum: "Fast-in-Slow: A Dual-System VLA Model Unifying Fast Manipulation within Slow Reasoning"
_NeurIPS.cc/2025/Conference — NeurIPS 2025 poster_

### Official Review · Reviewer_x4Fe · 2025-06-28

**Clarity:** 2
**Significance:** 2
**Originality:** 2
**Rating:** 4
**Confidence:** 4

**Summary:**

The work presents Fast-in-Slow (FiS-VLA), a dual-system vision-language-action model that repurposes the final transformer blocks of a pretrained VLM as a high-frequency execution module (System 1), while retaining the full model for slow reasoning (System 2). Key advantages include:
- Parameter sharing between System 1 and System 2 to leverage pretrained knowledge
- Asynchronous frequency design (1:4) for efficient control at 21.9 Hz
- Dual-aware co-training combining diffusion-based action generation and autoregressive prediction
- Integration of 3D point cloud inputs for precise manipulation

**Questions:**

1. **Parameter Reuse vs. New Layers like Fig. 1(a)**: Have you tested adding some new transformer blocks (instead of reusing VLM layers) with identical training settings? This would clarify if the architecture itself (not just parameter reuse) drives performance.
2. **Asynchronous Training Compatibility**: Can the asynchronous frequency ratio be applied to the separate-policy architecture in Figure 1(a)? If so, how does FiS-VLA's design uniquely benefit from this strategy?
3. **System 2's Role in Reasoning**: Please demonstrate how System 2's "slow reasoning" (e.g., task planning) specifically enhances System 1's actions causally, beyond just parameter sharing and co-training.
4. **RLBench Experimental Parity**: How do $\pi_0$ and CogACT's RLBench results compare to FiS-VLA in terms of:
   - Pretraining data (e.g., did they use the same 860K trajectories?)
   - Fine-tuning strategies (full vs. LoRA)
   - Input modalities (were 3D point clouds available to baselines?)
5. A recent study, namely \pi_{0.5}, also conducts language-action cotraining. Could you discuss the pros and cons between FiS-VLA and \pi_{0.5}?

**Ethical Concerns:**

["NO or VERY MINOR ethics concerns only"]

**Final Justification:**

The authors have adequately addressed my concerns. Therefore, I decided to increase my score.

**Limitations:**

Yes.

**Paper Formatting Concerns:**

Figures and Tables should be separated instead of being together in Tables 2 and 3.

**Quality:**

2

**Strengths And Weaknesses:**

## Strengths
1. **Clear Visualization**: Figures 1 and 2 effectively illustrate the architecture, highlighting how System 1 is embedded within the VLM, which aids reader comprehension.
2. **Fine-grained Experiments**: The study includes simulations on RLBench, real-world dual-arm tasks, ablation studies on modality inputs/frequency ratios, and generalization tests, providing robust validation.
3. **Performance Gains**: FiS-VLA outperforms SOTA by 8-11% in success rate and achieves 2× faster inference than baselines without action chunking.

## Weaknesses
### 1. Theoretical Alignment and Writing
- The core question/motivation ("If a VLM model serves as the ‘brain’ of the robot, can it integrate System 1 and System 2 processes to enable coordinated reasoning and execution?") is not convincing, since VLMs are primarily regarded as vision-language-focused instead of the whole 'brain'. And the discussion of Kahneman's dual-system theory feels superficial, as the so-called 'System 2' in FiS-VLA does not exhibit deliberate reasoning but serves as a feature extractor, which is not the "slow reasoning" from the aspect of System 2 in the cognitive field.

### 2. Architectural Innovation
- The approach mainly reuses existing VLM components (e.g., LLaMA2 blocks) rather than introducing novel structures. The key distinction lies in parameter sharing and training strategies, which may limit the architectural innovation.

### 3. Experimental Gaps
- **Parameter Reuse Validation**: The ablation on shared blocks (1-8) does not compare against adding new transformer layers (e.g., 2 new blocks vs. reusing 2 existing blocks), leaving uncertainty about whether performance gains stem from block reuse or not.
- **Asynchronous Training Novelty and Integrity**: Since the asynchronous training is not new in the field, as mentioned in line 39, the paper needs to show whether the proposed asynchronous training uniquely benefits FiS-VLA or can be applied to baseline architectures (e.g., Figure 1(a)).
- **Modal Input Necessity**: Removing 3D point clouds reduces performance to CogACT levels, suggesting the proposed dual-system design may not offer unique advantages under fair input conditions.
- **Co-training Mechanism**: Ablation 4 shows L_{slow} improves performance, but it is unclear whether this stems from preserving VLM reasoning or simply the discrete action classification loss.

---

> ### Author Rebuttal · Authors · 2025-07-31
>
> Dear Reviewer x4Fe,
>
> First of all, thanks for highlighting the clear visualization, fine-grained experiments, and performance gains of our work. We also sincerely appreciate your thoughtful comments and questions, which we address in detail below.
>
> ---
> ### **[W1 & W3.4 & Q3]. Clarification on Theoretical Alignment and System 2's Reasoning Capability.**
> **A. FiS-VLA draws inspiration from Kahneman’s dual-system theory not to replicate its full cognitive mechanisms, but to adopt its functional abstraction.** In this theory, System 1 is fast and automatic, while System 2 is slow, logical, and deliberative. This framework has been widely adopted as a functional abstraction in recent dual-system VLA models [1, 2, 3], where System 2 is implemented as a VLM. Specifically, GR00T N1 [1] treats VLM as a System 2 providing latent features for the diffusion head, while Hi Robot [2] generates low-level language commands. In both our approach and GR00T N1, System 2 goes beyond feature extraction, it performs multimodal fusion, semantic reasoning, and task understanding, supplying latent features that directly guide action generation, consistent with System 2’s deliberative role.
>
> Moreover, unlike prior training paradigms such as GR00T N1, we introduce an dual-aware co-training strategy to preserve System 2’s reasoning. In FiS-VLA, System 2 reasons about the obervations and task instructions, further generates discretized actions explicitly.
>
> **B. Evaluation of System 2's reasoning capability.** As shown in Lines 239–241, our System 2’s autoregressive supervision can leverage language-based task plans. To further assess whether preserving System 2’s reasoning improves FiS-VLA’s action generation, we replaced discrete action supervision with task plans automatically generated by Gemini 2.5 Pro and manually verified.
> For example, in the "phone_on_base" task from RLBench, the plan is “Step 1: Move the gripper close to the phone. Step 2: Grasp the phone. Step 3: Move it above the phone base. Step 4: Place it on the phone base.” We also use Gemini 2.5 Pro to augment plan diversity.
>
> As shown in the table below, FiS-VLA achieves a 73% average success rate with plan-based co-training, outperforming the 69% obtained using discrete actions. This suggests that explicit reasoning supervision leads to more accurate conditioning of System 1 and improved performance.
> ||Mean Success Rate|
> |---|---|
> |FiS-VLA (Task planning)|0.73|
> |FiS-VLA (Original)|0.69|
>
> We would like to discuss the insights behind this phenomenon. We find that preserving System 2’s reasoning supports deeper instruction comprehension and learning of richer contextual knowledge. This enables higher-quality conditioning for System 1. With shared parameters, System 1 inherits pretrained knowledge and better interprets latent features from System 2, thus improving robustness. We will integrate these findings in the revised version.
>
> **C. For the writing of "brain", it is intended as a metaphor.** It does not suggest that we have constructed a complete brain using VLM. The writing of Line 43-46 will be revised as follows: "Considering these limitations, and motivated by the functional abstraction of Kahneman's dual-system framework, we raise the following question: “If a VLM model serves as the central decision-making module of the robot, can it integrate System 1 and System 2 processes to enable coordinated reasoning and execution?"
>
> ---
> ### **[W2]. Architectural Innovation.**
> **We would like to clarify that the primary contribution of this work is proposing a new dual-system VLA paradigm.** Unlike prior works [1, 3] that rely on a separate execution model as System 1, which lacks internet-scale pretraining and introduces information loss at the boundary between the two systems (as validated in W3.1 and Q1), FiS-VLA is the first to repurpose the final layers of an LLM to serve as a fast execution module (System 1) embedded within a pretrained VLM (System 2). This tightly integrated design directly addresses the bottleneck of previous works, which, in our perspective, constitutes a nontrivial and novel dual-system paradiam contribution. Building on this core design, we conduct an in-depth investigation into how to fully exploit FiS-VLA's potential for coordinated dual-system operation, focusing specifically on three key aspects:
>
> 1. **The coordination of operating frequencies between the two systems**, exploring how to significantly improve the model's inference speed while maintaining manipulation accuracy and stability.
>
> 2. **The design of heterogeneous modality inputs for each system**, which efficiently embeds point cloud as observation conditions for the execution component (System 1), enhancing the model's spatial understanding in a simple yet effective manner.
>
> 3. **The co-optimization of System 1 and System 2**, strengthening their cooperation and improving overall task performance.
>
> ---
> ### **[W3.1 & Q1]. Parameter Reuse Validation.**
> To evaluate the effectiveness of our parameter-sharing design, we conducted the following ablation studies:
> + Experiment 1: The VLM serves as System 2, and the last two LLM layers are duplicated to form an independent System 1. This configuration supports both autoregressive generation from System 2's VLM and diffusion-based action prediction through the duplicated LLM layers. The entire system is trained using identical settings with FiS-VLA.
> + Experiment 2: This experiment mirrors the setup of Experiment 1, with the difference being that four LLM layers are duplicated instead of two.
>
> As shown in the table below, both variants underperform compared to FiS-VLA. We attribute this to feature misalignment. The LLM was pretrained across all 32 layers, enabling hierarchical (layer-by-layer) processing. Duplicating the last few layers breaks this chain (e.g., in Experiment 1, the new 31st layer in System 1 must process outputs from System 2’s 32nd layer), despite never being pretrained to do so. This disrupts feature compatibility and limits performance.
>
> In contrast, FiS-VLA’s parameter-sharing keeps the LLM’s original structure and information flow intact. System 1 remains embedded within System 2, preserving pretrained knowledge while enabling action execution. This design proves more efficient and effective than separated architectures.
> ||Mean Success Rate|
> |---|---|
> |Experiment 1 (Two independent layers)|0.61|
> |Experiment 2 (Four independent layers)|0.59|
> |FiS-VLA (Original)|0.69|
>
> ---
> ### **[W3.2 & Q2]. Asynchronous Training Compatibility.**
> **Our goal is not to simply use the asynchronous training method, but rather to explore an asynchronous training strategy that is optimally aligned with the new FiS-VLA dual-system paradigm.** The proposed asynchronous operation relies on the effective integration between System 2 and System 1, as System 1 must maintain high-performance execution with only low-frequency guidance from System 2.
>
> To validate the superiority of our proposed asynchronous training approach for FiS-VLA, we applied the same asynchronous strategy to CogACT, **using frequency ratios of 1:2 and 1:4**. As shown in the table below, CogACT's performance showed a significant decline. We find that naively applying an asynchronous frequency strategy to a VLA model can significantly reduce the informational context for the execution module during action generation. These results highlight the importance of FiS-VLA’s unified dual-system design and our dual-aware co-training strategy, both built with asynchronous frequency mechanism in mind.
> ||Mean Success Rate|
> |---|---|
> |CogACT (1:2)|0.53|
> |CogACT (1:4)|0.48|
> |CogACT (Original)|0.61|
>
> ---
> ### **[W3.3]. Modal Input Necessity.**
> **Heterogeneous modality input is a unique design of FiS-VLA.** Following your suggestion, we conducted additional experiments on CogACT under matched input and frequency settings. Specifically, we adopted the 1:4 asynchronous frequency to align CogACT’s System 1 updates with System 2. We also integrated our point cloud encoding into CogACT’s System 1. As shown in the table, FiS-VLA still outperforms CogACT, highlighting the advantage of our holistic dual-system design.
> || Mean Success Rate|
> |---|---|
> |CogACT (Original)|0.61|
> |CogACT (1:4 & System 1 + Point cloud)|0.54|
> |FiS-VLA (Original)|0.69|
>
> ---
> ### **[Q4]. RLBench Experimental Parity.**
> Thank you for your comments. We will include the RLBench experimental details in the revised version.
>
> + π₀ used both open-source and over 10,000 hours of self-collected data (\~10M frames) for pretraining. CogACT relied solely on open-source data (\~22.5M frames), comparable to our 860K videos (\~19M frames). As noted in Appendix Line 17, we followed CogACT's data processing.
>
> + FiS-VLA, π₀, and CogACT all used full fine-tuning for downstream tasks.
>
> + While FiS-VLA uniquely uses heterogeneous modality input, other baselines can also incorporate point cloud data. For example, CogACT's results are reported in W3.3.
>
> ---
> ### **[Q5]. FiS-VLA and π₀.₅**
> We consider π₀.₅ a valuable work that emphasizes the importance of data, leveraging large-scale self-collected robotic and internet-sourced data to boost generalization. However, our approach differs in both implementation objectives and strategies. π₀.₅ extends π₀’s architecture by adding an action expert (System 1) after the VLM (System 2) and using additional training data to equip the VLM with task planning capabilities. In contrast, FiS-VLA focuses on a new and unified dual-system paradigm designed to achieve more effective and lossless coordination between System 2 and System 1.
>
> ---
> ### **Paper Formatting Concerns**
> Thank you for your suggestion. We will separate the figures and tables in the revised version.
>
> ---
> [1] GR00T N1: An Open Foundation Model for Generalist Humanoid Robots
>
> [2] Hi Robot: Open-Ended Instruction Following with Hierarchical Vision-Language-Action Models
>
> [3] Towards Synergistic, Generalized, and Efficient Dual-System for Robotic Manipulation

---

> > ### Author Response · Authors · 2025-08-03
> > **Further discussion to Reviewer x4Fe**
> >
> > Dear Reviewer x4Fe,
> >
> > First of all, thank you very much for taking the time to review our paper and for your valuable suggestions. We have taken your concerns seriously and provided detailed responses. Specifically, we have provided detailed explanations and corresponding empirical validations regarding both the reasoning capability of FiS-VLA’s System 2 and the architectural innovations of FiS-VLA.
> > Meanwhile, we have included additional experiments and analysis to demonstrate the effectiveness of FiS-VLA’s parameter reuse strategy, its asynchronous frequency design, and the handling of heterogeneous modality inputs, along with discussions on the motivations behind these designs. We will incorporate the updates from this rebuttal into the revised version of the paper to further enhance the completeness of our work.
> >
> > Therefore, we hope our responses have addressed your concerns, and we welcome any further discussion or questions you might have. If you feel that our clarifications have addressed your concerns, we would greatly appreciate your consideration in revisiting your rating, as a higher score would serve as a meaningful recognition of our efforts.
> >
> > Paper 4688 authors

---

> ### Comment · Reviewer_x4Fe · 2025-08-04
>
> Thank you for your detailed reply.
>
> After reviewing it, I still feel that there is insufficient evidence to strongly support the extensive association drawn between FiS-VLA and the Dual-process theory, particularly regarding the role of System 2 reasoning and its specific function within the FiS-VLA model.​ Accordingly, I would suggest that in addition to including the promised supplementary experiments, the writing should adopt a more conservative approach when integrating FiS-VLA with the Dual-process theory. This would enhance the rigor of the paper, help avoid potential controversies.
>
> Regarding the supplementary experiments on asynchronous training compatibility (ATC), could you provide more in-depth analysis? For instance, why does the introduction of ATC lead to poorer performance in CogACT, and what specific differences between CogACT and FiS-VLA might account for this discrepancy?​
>
> Additionally, in the experiments on modal input necessity, I would appreciate seeing the results for CogACT + Point cloud. This would allow for a more direct comparison of the performance gaps between FiS-VLA and CogACT in scenarios with and without point cloud input.
>
> I would be willing to revise my evaluation positively once the remaining issues are properly addressed.

---

> > ### Author Response · Authors · 2025-08-05
> > **Further discussion to Reviewer x4Fe**
> >
> > ### **[Point-B]. More in-depth analysis with asynchronous training compatibility.**
> > Regarding why CogACT performs worse under asynchronous control frequencies, we would like to discuss two main reasons for this, and for each point, we will also explain why FiS-VLA outperforms CogACT:
> >
> > 1. **Difference of conditioning manner between System 1 and System 2.** In CogACT, the output of System 2 is compressed into **a single “cognition token”**, which is then used as the condition for System 1’s action generation. While this design is highly effective in the original CogACT framework for improving the efficiency of action module generation, it becomes somewhat limiting under asynchronous frequency settings. In such cases, System 1 in CogACT can only receive the “observation and language instruction” feature (single cognition token) produced by System 2 at a relatively lower frequency, which leads to a significant reduction in conditioning information, even though these multimodal conditions are critical for guiding action generation. In contrast, FiS-VLA retains the **full token representation** of System 2’s multimodal understanding of “observations and language instructions” as the condition for System 1. Meanwhile, the VLM (System 2) is pretrained on internet-scale data using the full 32-layer Transformer architecture for forward feature propagation. FiS-VLA shares the final few Transformer blocks as System 1, enabling it to better interpret earlier-layer features without compromising the integrity of the pretrained VLM. Therefore, FiS-VLA enables more effective processing and utilization of conditioning features passed from the preceding layers, ensuring better cooperation between the two systems.
> > 2. **Differences in architecture and training design.** CogACT’s System 1 is a relatively **small DiT model, pretrained only on robot trajectory data**. As a result, it lacks exposure to common-sense image-instruction feature learning gained from internet-scale pretraining. Furthermore, the limited model size and training data constrain its action head capacity, reducing its ability to learn from new asynchronous and multimodal inputs encountered in downstream tasks. In contrast, FiS-VLA’s System 1 directly reuses the **final Transformer blocks of a pretrained LLM**. This design allows **System 1 to naturally inherit the LLM’s internet-scale pretraining knowledge**, thereby enhancing the execution component capacity of FiS-VLA to effectively learn to use new asynchronous and multimodal inputs as conditions for action generation.

---

> > ### Author Response · Authors · 2025-08-05
> > **Further discussion to Reviewer x4Fe**
> >
> > ### **[Point-C]. Experiments of adding point cloud to CogACT.**
> > Following your valuable suggestion, in addition to the supplementary experiments in rebuttal W3.3, we also incorporate point cloud data into CogACT (Original). Specifically, we conduct two experiments: (1) **adding point cloud data before the entire VLA model**, and (2) **before the diffusion action head**. The results are summarized in the table below. When point cloud data is added only before entire CogACT model, the performance is similar to that of CogACT (Original). When added only to the action head, the performance improves over the original version, but there is still a 5-point drop compared to FiS-VLA (Original). Below is our analysis based on the experimental results and corresponding observations.
> >
> > |  | Mean Success Rate |
> > | --- | --- |
> > | CogACT (System 2 + Point cloud) | 0.62 |
> > | CogACT (System 1 + Point cloud) | 0.64 |
> > | CogACT (Original) | 0.61 |
> > | FiS-VLA (Original) | 0.69 |
> > | FiS-VLA (Task planning) | 0.73 |
> >
> >
> > Although the specific cognition token design is highly effective in the original CogACT framework, there are two potential limitations regarding the understanding of point cloud data.
> >
> > 1. First, as analyzed in Point-B.1, a single cognition token is sufficient for representing 2D visual and language understanding due to the model's large-scale robot pretraining. However, it is **inadequate for representing newly injected point cloud data in downstream tasks**. This does not imply a flaw in the cognition token design itself but rather reflects the fact that most real-world robot pretraining data does not include point cloud information.
> > 2. Second, using our proposed point cloud injection method, we directly inject point cloud tokens as conditioning inputs before CogACT’s action component. This leads to performance gains compared to CogACT (Original). The smaller improvement observed in CogACT may result from the **imbalance in token quantities**, as newly injected conditioning tokens significantly outnumber cognition token. Additionally, as analyzed in Point-B.2, the **model capacity of the action component** also limits the effectiveness of newly injected conditioning tokens integration.
> >
> > ---
> >
> > Finally, thank you once again for your constructive suggestions, which have made our paper more rigorous and comprehensive. We will incorporate all the revisions you suggested, including more conservative descriptions, additional experiments, and corresponding in-depth analyses in the revised version. We hope our further responses have addressed your concerns. If you feel that our clarifications are satisfactory, we would sincerely appreciate your consideration in updating your evaluation positively. Of course, if there are still any unresolved concerns, we would be glad to continue the discussion and further improve the work.

---

> > ### Author Response · Authors · 2025-08-06
> > **Further discussion to Reviewer x4Fe**
> >
> > Dear Reviewer x4Fe,
> >
> > As the discussion period comes to a close, we sincerely appreciate your time and valuable feedback. We apologize for the interruption and would like to confirm whether our additional responses have adequately addressed your concerns.
> >
> > Specifically, we have: 1. Revised the manuscript to clarify potential misunderstandings and ensure a more balanced presentation. 2. Provided a deeper analysis of asynchronous training compatibility, further strengthening our methodological contributions. 3. Supplemented the experimental results by integrating point cloud information directly into CogACT (Original), offering a more comprehensive evaluation.
> >
> > Thank you once again for your valuable feedback and time.
> >
> > Paper 4688 authors

---

> > > ### Comment · Reviewer_x4Fe · 2025-08-07
> > >
> > > Thank you for your reply. I believe the new experiments you have added enhance the novelty and appeal of this work, and they perform well on the issues I care about. To my understanding, this work features designs and experimental results that could inspire future research. Therefore, on the condition that you commit to updating the revised version with the newly added textual descriptions and experimental content, I will raise my evaluation score.

---

> > > > ### Author Response · Authors · 2025-08-07
> > > > **Further discussion to Reviewer x4Fe**
> > > >
> > > > Dear Reviewer x4Fe,
> > > >
> > > > We are truly delighted to have addressed your questions and concerns, and we sincerely appreciate your willingness to raise your evaluation score. We will incorporate all the additional textual descriptions and experimental results from this rebuttal into the revised version of our paper, as we believe this exchange has further enhanced the completeness and rigor of our work. Finally, thank you once again for your valuable comments and time.
> > > >
> > > > Paper 4688 authors

---

> ### Author Response · Authors · 2025-08-05
> **Further discussion to Reviewer x4Fe**
>
> Dear Reviewer x4Fe,
>
> First of all, we sincerely appreciate your constructive feedback on our rebuttal. We will address each of your points in detail.
>
> ---
>
> ### **[Point-A]. More conservative writing.**
> We sincerely apologize for any confusion our original writing may have caused. We will revise the descriptions regarding the connection to Dual-process theory more carefully, rather than simply following prior works. In the revised version, we will rewrite **Lines 43–46** as follows:
>
> > Considering these limitations, **and in order to improve VLA execution efficiency while maintaining its inherent capabilities, we raise the following question: If a VLM model serves as the central decision-making module of the robot, can it integrate System 2 and System 1 processes to enable coordinated multimodal comprehension and fast execution?**
>
> In addition, as you suggested, we will revise the usage of the term “reasoning” by providing a more precise description of System 2’s specific role within the FiS-VLA model, such as multimodal comprehension, providing latent multimodal conditions, and generating discrete action or language planning.
>
> For instance, we will modify **Lines 4–7 and 12–14** of the main paper as follows:
>
> > To mitigate this dilemma, dual-system approaches have been proposed to **leverage a VLM-based System 2 module for handling high-level multimodal comprehension**, and a separate System 1 action module for ensuring real-time control.
>
> > This innovative paradigm not only enables high-frequency execution in System 1, **but also facilitates coordination between multimodal comprehension and execution components** within a single foundation model of System2.
>
> Meanwhile, we will revise **Lines 17–20, 47–49, and 66-68** as follows:
>
> > To enable coordination between the two systems, a dual-aware co-training strategy is proposed that equips System 1 with action generation capabilities while **preserving System 2’s contextual understanding to provide stable latent conditions for System 1**.
>
> > To this end, we propose Fast-in-Slow (FiS), a VLA foundation model that integrates the fast execution capabilities of System 1 into a pretrained VLM, **while preserving its inherent System 2 multimodal understanding and generation capabilities**.
>
> > **For the multimodal comprehension component (System 2)**, we exploit an autoregressive next-token prediction objective to **maintain its discrete action generation or high-level language planning capabilities and preserve the overall coherence and integrity of System 2**.
>
> We greatly appreciate your thoughtful suggestions. In the revised version, we will not only include the promised planning experiments, but also adopt a more conservative and detailed writing to avoid all potential controversies. Indeed, we aim to propose a new dual-system VLA paradigm to the embodied intelligence community while also providing a more conservative and rigorous writing style for future work to follow.

---

### Official Review · Reviewer_y6xs · 2025-06-29

**Clarity:** 3
**Significance:** 3
**Originality:** 3
**Rating:** 6
**Confidence:** 4

**Summary:**

This paper propose a novel duel-system Fis-vla model which unifies system 1 and system 2 into a single pretrained vlm model without introducing new parameters. Experiments in both simulation and real-world demonstrate that this Fis-VLA model not only reaches much higher control frequency but also higher task success rates.

**Questions:**

1. Could the author explain why Fis-vla can achieve higher success rate even in static tasks which not requires high control frequency? Can you provide some examples or intuitions?

2. What about also put point cloud into the beginning of the VLM model?

**Ethical Concerns:**

["NO or VERY MINOR ethics concerns only"]

**Final Justification:**

After rebuttal, all my concerns have been resolved and I continue to recommend acceptance. And I also read the discusions between author and other reviewers which further confirm my acceptance.

**Quality:**

3

**Strengths And Weaknesses:**

Strengths:

1. This paper proposes an elegant approach to implementing a dual-system VLA model without introducing additional parameters, which I find quite reasonable. In contrast, previous works often use untrained action heads or action experts, which may compromise the pretrained weights in the vision-language model (VLM).

2. The method’s effectiveness is validated through experiments in both simulation and real-world settings. Ablation studies further confirm the contribution of each component.

Weakness:

1. The advantages of a high-frequency policy could be more evident in tasks involving dynamic changes or requiring fine-grained dexterity. The current evaluation focuses primarily on pick-and-place tasks, which may not fully demonstrate the benefits of increased policy frequency.

---

> ### Author Rebuttal · Authors · 2025-07-31
>
> Dear Reviewer y6xs,
>
> First of all, thanks for describing our FiS-VLA as an elegant approach and quite reasonable. We also sincerely appreciate your thoughtful comments and questions, which we address in detail below.
>
> ---
>
> ### **[W1]. Additional dynamic change experiments to further evaluate our method.**
>
> Thank you for your suggestion. We have added a new set of experimental scenarios involving dynamic changes to the manipulated objects. This experiment further demonstrates our model’s response frequency and closed-loop manipulation capability. The task is "grasp wine bottles and place them on the rack," as shown in Table 2 of the main paper and Figure 9 of the Appendix. During task execution, we manually move the bottles in four directions: up, down, left, and right. The setup of this real-world experiment is consistent with that described in Section 4.3 of the main paper.
>
> We evaluated 20 rollouts, each using a different approach to movement. The mean success rates of FiS-VLA and π₀ on this task are reported in the table below. In this task, our model continuously and smoothly adjusts the manipulation pose in response to the moving object, achieving satisfactory manipulation results. In the revised version, we will also include more specially designed tasks that better highlight the advantages of our fast-slow system design.
>
> | | Place bottles at rack (Original) | Place bottles at rack (Dynamic changes) |
> | --- | --- | --- |
> | FiS-VLA |0.70|0.65|
> | π₀ |0.55|0.40|
>
>
> ---
>
>
> ### **[Q1]. Why can FiS-VLA achieve a higher success rate in static tasks?**
>
> Thank you for your insightful question. Our model not only achieves high-frequency action generation capability but also contains a series of robotic-specific designs that contribute to its strong performance across diverse tasks:
>
> 1. We innovatively integrate System 1 into the VLM (System 2) and adopt a dual-aware co-training strategy to achieve deep fusion between semantic understanding and action generation. This enables the construction of more robust action representations, allowing the model to adapt to various task types, from static operations to complex manipulation tasks. For example, our method can handle both precise pick-and-place tasks and relatively complex tasks such as "_wiping a blackboard"_ and "_folding a deformable towel and placing it into a basket"_.
>
> 2. In addition, our proposed heterogeneous modality input helps System 1 dynamically and efficiently acquire 2D images, 3D point clouds, and robot states at each timestep. Even for static tasks, this provides richer perceptual information and improves action precision.
>
> 3. Finally, our asynchronous frequency design enables high-frequency robot control, generating smoother and more stable control trajectories, and avoiding issues such as object slippage or inaccurate positioning caused by jerky motion.
>
> ---
>
> ### **[Q2]. The experiment of feeding point clouds into the beginning of the VLM.**
>
> This is an important ablation study explored in our work, and we have included the experimental results in the Appendix. As shown in Section B.2 and Figure 2 of the Appendix, we investigate various combinations of heterogeneous modality inputs for the fast and slow systems. Specifically, **Variant 1 feeds the point cloud directly into the beginning of the VLM model**, together with the 2D image and language instruction as input. The corresponding results are shown in the table below.
>
> | | Mean Success Rate |
> | --- | --- |
> | FiS-VLA | 0.69 |
> | Variant 1 | 0.63 |
>
> The results demonstrate that using the point cloud solely as input to the slow system does not improve manipulation accuracy, while requiring additional training and inference time. In order to maintain both model accuracy and speed, we only introduce the 3D point cloud as a condition for the execution component (System 1).

---

> > ### Comment · Reviewer_y6xs · 2025-08-03
> >
> > Thank you for your rebuttal response to my questions and the additional experiments you provided, which have addressed my main concerns. I suggest that the authors include the dynamic changes experiment and the exploration of point cloud inputs in the main paper. Based on the authors’ rebuttal and the comments from other reviewers, I am willing to improve my rating and continue to recommend acceptance.

---

> > > ### Author Response · Authors · 2025-08-03
> > > **Further discussion to Reviewer y6xs**
> > >
> > > Dear Reviewer y6xs,
> > >
> > > Thank you very much for your recognition of our work. We will include the dynamic changes experiment in Section 4.3 of the main paper (Real-World Experiment), and the discussion on point clouds will be added to Section 4.2 (Ablation Study) of the main paper.
> > >
> > > Paper 4688 authors

---

> ### Author Response · Authors · 2025-08-03
> **Further discussion to Reviewer y6xs**
>
> Dear Reviewer y6xs,
>
> First and foremost, we sincerely appreciate the time and effort you dedicated to reviewing our paper, as well as your valuable suggestions and recognition of our work. To further demonstrate the superiority of FiS-VLA's high-frequency policy, we have incorporated additional task featuring dynamic changes. Additionally, we provide insights into why FiS-VLA performs well in both static and dynamic tasks, along with further clarification regarding the point cloud input concerns.
>
> Therefore, we hope our rebuttal has addressed your concerns, and would be truly grateful for your consideration in updating the score if you find our responses satisfactory. If any questions remain, we would be more than happy to discuss and provide a prompt response. Thank you again for your valuable feedback.
>
> Paper 4688 authors

---

### Official Review · Reviewer_qC7A · 2025-07-01

**Clarity:** 4
**Significance:** 3
**Originality:** 2
**Rating:** 4
**Confidence:** 3

**Summary:**

The paper aims to improve execution efficiency for Vision-Langage-Action (VLA) models for robotics manipulation.
It proposes Fast-in-Slow (FiS), a unified dual-system VLA model that embeds the System-1 execution module within the VLM-based System-2 by partially sharing parameters, instead of a separate System-1 policy head.
The two systems are designed to incorporate heterogeneous modality inputs and asynchronous operating frequencies, enabling both fast and precise manipulation
Under this paradigm, FiS-VLA achieves superior performance and high-frequency control.

**Questions:**

π0 employs a 2.6B-parameter LLM, while FiS-VLA is based on a 7B-parameter LLM. Thus, is the success rate in Tab. 2 a fair comparison?

**Ethical Concerns:**

["NO or VERY MINOR ethics concerns only"]

**Final Justification:**

I thank the authors for their response. My main concerns have been addressed. I will keep my original rating.

**Limitations:**

The authors discuss the limitations.
As is mentioned in Sec. 5, it is promising to enable dynamic adaptation of the shared parameters and the collaboration frequency between the two systems.

**Quality:**

3

**Strengths And Weaknesses:**

### Strengths
1. The method is sound and reasonable.
- Existing methods usually introduce a new System-1 policy head, while FiS-VLA integrates a System-1 execution module into a System-2 VLM model, leveraging shared transformer blocks. Thus, FiS-VLA fully leverages the VLM’s pre-trained knowledge and reasoning capabilities and also enables high-frequency control with an asynchronous architecture.
- Furthermore, System-1 and System-2 are provided with heterogeneous input modalities specifically tailored to their functions. Since the System-2 VLM receives language instructions and 2D visual observations to match internet-scale pretraining on image-text paired data. In contrast, System-1 processes 3D point clouds and robot states to produce stable and responsive actions.

2. Extensive experiments have been conducted.
FiS-VLA has been compared in both the simulation environment RLBench and real-world scenarios. For real-world verification, the authors evaluate four tasks on the Agilex Robot and AlphaBot, where FiS-VLA shows better generalization.


### Weaknesses
Execution efficiency is not compared in real-world scenarios. Fig. 1 and Tab. 1 have shown inference speed compared on RLBench. Inference speed improvement has not been shown, e.g., task progress synchronization in real-world video demos.

---

> ### Author Rebuttal · Authors · 2025-07-30
>
> Dear Reviewer qC7A,
>
> First of all, thanks for recognizing that our method is sound and reasonable. We also sincerely appreciate your thoughtful comments and questions, which we address in detail below.
>
> ---
>
> ### **[W1]. The comparison of execution efficiency in real-world scenarios.**
>
> The model inference speed is similar in both real-world and simulation environments, as we deploy the model on an NVIDIA 4090 GPU for both. However, real-world deployment introduces additional latency due to hardware and communication overhead. Under the same hardware setup (Agilex dual-arm robot, as shown in Figure 1 of the appendix) and with an action chunk size of one, the control frequency comparison between our method and the baselines is summarized in the table below.
>
> | | Control Frequency       |
> |------------|--------------------------|
> | CogACT     | 7.1 Hz |
> | π₀         | 9.2 Hz |
> | FiS-VLA    | 11.6 Hz |
>
>
> For the detailed latency in real-world control, the baseline models take approximately 34 ms to obtain 2D observations, whereas our method requires 40 ms to acquire 2D + 3D data. Meanwhile, there are some unavoidable sources of latency, including communication between the edge device and the robot, the robot’s actuation time, and the inverse kinematics computation used to convert the end-effector pose into joint positions (depending on the control paradigm), which together contribute around 1–2 ms. The majority of the latency comes from model inference: our model takes approximately 45 ms per forward pass, whereas π₀ requires around 72 ms per pass.
>
> To improve control frequency in real-world scenarios, we incorporate an action chunking strategy into the FiS-VLA framework, as described in Section B.1 and shown in Figure 2 of the Appendix. For example, with a chunk size of 8, the model achieves a satisfactory manipulation success rate, while the control frequency on the real robot can exceed 20 Hz. This enables real-time control even when accounting for additional latency caused by hardware and communication overhead.
>
> Finally, due to rebuttal constraints that prohibit the inclusion of visualizations and videos, we will provide execution video demos comparing other baselines with FiS-VLA in the revised supplementary material.
>
> ---
>
> ### **[Q1]. Explanation of the experiments in Table 2 of main paper.**
>
> Thank you for your constructive question. First, we would like to clarify that the comparison between our 7B version of FiS-VLA and π₀ is fair, for the following reasons:
>
> 1. Both models were trained on the same amount of downstream robotic data and employed a full-parameter fine-tuning strategy, ensuring a fair training setup.
> 2. FiS-VLA outperforms π₀ in both performance and speed, and our comparison is conducted along these two dimensions, which demonstrates the fairness of our evaluation.
> 3. Due to our unique dual-system asynchronous frequency design, the actual calling frequency of the entire LLM in our framework is lower than that of π₀.
>
> To further address your question, we conducted an additional comparison using the 2.7B Phi-2 model as the LLM backbone. To ensure fairness, FiS-VLA (2.7B) was also pretrained on the same assembled robotic datasets under identical settings. Due to the time constraints of the rebuttal period, we only trained for one epoch. As shown in the table below, the compact-sized FiS-VLA also outperforms π₀ in performance, which demonstrates the effectiveness of our approach and its generalizability to different VLM backbones.
>
> | | Mean Success Rate |
> | --- | --- |
> | FiS-VLA (2.7B) | 0.62 |
> | π₀ | 0.55 |

---

> > ### Author Response · Authors · 2025-08-03
> > **Further discussion to Reviewer qC7A**
> >
> > Dear Reviewer qC7A,
> >
> > We are truly grateful for the time and effort you put into reviewing our work. To address your concerns regarding the real-world deployment of our model compared to the baselines, we report the control frequencies and actual runtime of FiS-VLA and the baseline models under real-world scenario. To further ensure fair comparison with π0, we also evaluated FiS-VLA using a smaller 2.7B LLM backbone.
> >
> > Therefore, we hope these responses have adequately addressed your comments, and we remain available for any further discussion. If our rebuttal has resolved your concerns, we would greatly appreciate your consideration of a higher rating, as it would be a strong encouragement for our work.
> >
> > Paper 4688 authors

---

> > ### Comment · Reviewer_qC7A · 2025-08-05
> >
> > I thank the authors for their detailed response.
> > My main concerns have been addressed.
> >
> > I suggest that the authors include the additional analysis about real-world execution efficiency and the additional performance comparison in the main paper.
> >
> > I will keep my original rating.

---

> > > ### Author Response · Authors · 2025-08-07
> > >
> > > Dear Reviewer qC7A,
> > >
> > > We are very pleased to have addressed your main concerns. We will incorporate the additional analysis on real-world execution efficiency and the extended performance comparison into the main paper. Thank you once again for your valuable comments and time.
> > >
> > > Paper 4688 authors

---

### Official Review · Reviewer_Vt2R · 2025-07-05

**Clarity:** 3
**Significance:** 3
**Originality:** 3
**Rating:** 4
**Confidence:** 4

**Summary:**

This paper proposes a dual-system VLA model for robot manipulation, dubbed as FiS-VLA, which embeds a fast model (System 1 for control) and a slow model (System 2 for planning) within a same large vision-language model, where the two systems operate asynchronously to achieve both planning effectiveness and execution efficiency. Experiments demonstrate that FiS-VLA surpasses CogACT and pi0 in success rate and inference speed.

**Questions:**

-	What’s the bottleneck to real-world deployment? The expected latency per step to a ~20 Hz control system is ~50ms. To my knowledge, the time cost by camera capture (rgb-d and point cloud), communication, IK and other components could not be ignored at this scale.

**Ethical Concerns:**

["NO or VERY MINOR ethics concerns only"]

**Final Justification:**

The authors address all of my concerns, including the necessity of sharing VLM layers, comparison fairness, and execution latency. I believe that the results of this exchange will make this paper technically sounder.
Since I’ve already recommended borderline accept, I decide to keep my original rating.

**Limitations:**

Yes.

**Paper Formatting Concerns:**

None observed.

**Quality:**

3

**Strengths And Weaknesses:**

Strength:
-	This paper is well-written and the overall presentation is easy to follow.
-	The proposed fast-in-slow architecture is novel and promising. Sharing the last layers of VLMs as a fast-system for high-frequency execution is intuitive.

Weakness:
-	Lack detailed discussion on the necessity of sharing VLM layers. For example, how about independently training the two systems if we just copy the last layers as an independent model?
-	FiS-VLM is only tested on Llama2-7B, while choosing base models like Llama-3.2-1B/3B could guarantee a fairer comparison to pi0.

---

> ### Author Rebuttal · Authors · 2025-07-30
>
> Dear Reviewer Vt2R,
>
> First of all, thanks for recognizing that the fast-in-slow architecture is novel and promising. We also sincerely appreciate your thoughtful comments and questions, which we address in detail below.
>
> ---
>
> ### **[W1]. Why is sharing VLM layers necessary?**
> We sincerely appreciate your suggestion, which prompted us to provide further discussion on the necessity of sharing VLM layers. In our FiS-VLA framework, we unify System 2 and System 1 within a single VLM architecture by sharing the last few Transformer blocks. To systematically validate the advantages of this parameter-sharing approach, we conducted the following ablation studies:
>
> + Experiment 1: The VLM serves as System 2, while the last two layers of the LLM are copied to form a diffusion head after the VLM, which serves as an independent System 1. This configuration enables: (1) autoregressive generation through the VLM's outputs, and (2) diffusion-based action generation via the duplicated Transformer blocks. The entire model is jointly optimized using our proposed dual-aware co-training strategy.
> + Experiment 2: The experimental setup remains largely the same as in Experiment 1, except that the two copied Transformer blocks in System 1 are replaced with four blocks.
>
> As shown in the table below, the results demonstrate that simply replicating additional Transformer blocks yields inferior performance compared to FiS-VLA's shared-parameter design. We attribute this to the fact that the VLM is pretrained on internet-scale data using the full 32-layer Transformer architecture for forward feature propagation. When System 1 is implemented as an independent head composed of copied 31st and 32nd layers, it inherits pretrained weights but lacks the context in which those layers were originally trained. Specifically, the duplicated 31st layer in System 1 must process latent features from System 2’s 32nd layer, an input distribution it was never exposed to during pretraining. In contrast, the FiS-VLA framework shares the last few Transformer blocks as System 1, allowing for better interpretation of earlier-layer features without compromising the integrity of the pretrained VLM. This newly proposed paradigm, which embeds System 1 within System 2, enables lossless utilization of the VLM’s pretrained knowledge and facilitates more efficient model learning.
>
> | | Mean Success Rate |
> | --- | --- |
> | Experiment 1 (Two independent layers) | 0.61 |
> | Experiment 2 (Four independent layers) | 0.59 |
> | FiS-VLA (Original) | 0.69 |
>
>
> ---
>
>
> ### **[W2]. An additional comparision with π₀ using 2.7B LLM.**
>
> First, following your suggestion, we adopt the 2.7B Phi-2 model as our LLM backbone to not only further validate the effectiveness of our approach, but also demonstrate its generalizability to different LLM architectures. Given the critical role of large-scale robotic pretraining in achieving robust downstream performance for VLA tasks, we conduct the same pretraining on our assembled robotic datasets. Due to the time constraints of the rebuttal period, we only trained for one epoch. As shown in the table below, the results indicate that our lightweight FiS-VLA implementation still outperforms π₀.
>
> | | Mean Success Rate |
> | --- | --- |
> | FiS-VLA (2.7B) | 0.62 |
> | π₀ | 0.55 |
>
> Second, we would like to clarify that the comparison between our 7B version of FiS-VLA and π₀ is fair. Both models underwent downstream training on the same dataset, ensuring fair training conditions. Additionally, FiS-VLA outperforms π₀ in both performance and speed, and our comparison across these two metrics aims to support the fairness of our evaluation.
>
>
> ---
>
>
> ### **[Q1]. The control frequency comparison in real-world deployment.**
>
> First of all, thank you for raising this highly practical and important question. In the main paper, we primarily report speed-related quantitative results in the simulator, where all comparisons are conducted fairly. During real-world deployment with an action chunk size of one, FiS-VLA achieves a control frequency of 11.6 Hz, compared to 9.2 Hz for π₀ and 7.1 Hz for CogACT.
>
> Next, we outline the main sources of latency in real-world deployment. The baseline model takes approximately 34 ms to obtain 2D observations, whereas our method requires 40 ms to acquire 2D + 3D data. The additional overhead primarily arises from communication between the edge device (NVIDIA 4090 GPU) and the robot, the robot’s actuation time, and the inverse kinematics (IK) computation to convert the end-effector pose into joint position, together contributing around 1 to 2 ms. The dominant latency source is model inference, which takes approximately 45 ms per step.
>
> Meanwhile, to further improve control frequency in the real world, we adopt an action chunking scheme, as shown in Section B.1 and Figure 2 of Appendix. For example, when the chunk size is set to 8, the model accuracy remains nearly unchanged, but the real-world control frequency can exceed 20 Hz. This enables real-time control even when accounting for additional latency.

---

> > ### Author Response · Authors · 2025-08-03
> > **Further discussion to Reviewer Vt2R**
> >
> > Dear Reviewer Vt2R,
> >
> > We sincerely thank you for your time and thoughtful suggestions. We provided detailed explanations and supporting experimental validations to address: 1) the effectiveness of parameter sharing in System 1 of FiS-VLA, 2) the fairness of LLM backbone size comparisons across baseline models, and 3) the practical deployment bottlenecks of FiS-VLA in real-world scenarios.
> >
> > Therefore, we sincerely hope that our rebuttal has addressed your concerns. If so, we would greatly appreciate your consideration of raising the score, which would be a meaningful affirmation of our efforts. If you have any remaining questions, we would be happy to discuss and address them. Thank you once again for your valuable feedback.
> >
> > Paper 4688 authors

---

> > ### Comment · Reviewer_Vt2R · 2025-08-05
> > **The authors address all of my concerns and I will keep my original positive rating.**
> >
> > The authors address all of my concerns, including the necessity of sharing VLM layers, comparison fairness, and execution latency. I believe that the results of this exchange will make this paper technically sounder. Since I’ve already recommended borderline accept, I decide to keep my original rating.

---

> > > ### Author Response · Authors · 2025-08-06
> > > **Response to Reviewer Vt2R**
> > >
> > > Dear Reviewer Vt2R,
> > >
> > > We are truly glad to have addressed all of your concerns, and we sincerely thank you for your thoughtful suggestions, which have helped make our paper technically sounder. **We greatly appreciate your positive feedback and rating, and we will incorporate the insights from this rebuttal into the revised version of our paper.**
> > >
> > > Paper 4688 authors

---

### Note · Authors · 2025-08-15

Dear AC and Reviewers,

We sincerely thank you for your time, thoughtful feedback, and constructive discussions, which have been instrumental in strengthening our paper.
### **1. Strengths Highlighted by Reviewers**
We are encouraged that the novelty and soundness of our fast-in-slow VLA framework were widely recognized. Reviewer Vt2R noted that our architecture **is novel and promising**. Reviewer qC7A considered our method **sound and reasonable**, while Reviewer y6xs praised it as **an elegant approach** and **quite reasonable** for implementing a dual-system VLA without introducing additional parameters. Reviewer x4Fe further commended our work for its **clear visualization**, **fine-grained experiments**, and **performance gain**.
### **2. Our Responses and discussions**
During the discussion phase, we were honored to address all reviewer concerns. Specifically, we assessed the shared-parameter design and evaluated performance with a smaller LLM backbone. Reviewer Vt2R and Reviewer qC7A noted, "**The authors address all of my concerns**" and "**My main concerns have been addressed**", respectively.

Meanwhile, we tested adaptability in dynamically changing tasks for Reviewer y6xs, who responded, "**which have addressed my main concerns**" and "**I am willing to improve my rating and continue to recommend acceptance.**"

Following Reviewer x4Fe's comments, we integrated language planning capabilities and compared the baseline's asynchronous frequencies and point cloud inputs against FiS-VLA. Reviewer x4Fe remarked, "**They perform well on the issues I care about**" and "**I will raise my evaluation score.**"
### **3. Closing Statement**
We believe this work represents an important step toward building more reliable and efficient VLA models. FiS-VLA proposes a new dual-system VLA paradigm that embeds System 1 execution within a pretrained VLM while retaining its System 2 pretrained knowledge, enabling better coordination. It is the first to combine asynchronous operating frequencies with heterogeneous modality inputs, achieving both rapid execution and precise manipulation. FiS-VLA attains SOTA performance in simulation and real-world settings, with an inference speed of 117.7 Hz on an NVIDIA 4090 GPU (action chunk size = 8).

Finally, as the rebuttal has addressed all concerns and received positive ratings, we express our gratitude to the AC and reviewers for their constructive engagement, and look forward to a favorable consideration.

---

### Decision · Program_Chairs · 2025-09-17

**Decision:**

Accept (poster)

**Comment:**

This paper presents a dual-system model for robot manipulation that unifies a fast execution module and a slow reasoning module within a single model - repurposing the final layers of the language model for high-frequency control. The model outperforms prior methods in both simulation and real-world tasks. The key strength of this work is its simple and effective architecture. The paper is supported by a thorough set of experiments, including real-world demonstrations and detailed ablation studies that validate the contributions of the different design choices.

The author-reviewer discussion was constructive and improved the paper. Initially, some of the experimental comparisons and design justifications were not clear, but these issues were mainly resolved during the discussion. Reviewers raised critical questions about the necessity of parameter sharing versus adding new layers (Vt2R, x4Fe), the fairness of comparisons to baselines using smaller models (Vt2R, qC7A), and the need for experiments in dynamic environments to better highlight the method's advantages (y6xs). Reviewers confirmed that their concerns were addressed, leading to an improved consensus.